# Towards a Circular Economy of Plastics: An Evaluation of the Systematic Transition to a New Generation of Bioplastics

**DOI:** 10.3390/polym14061203

**Published:** 2022-03-17

**Authors:** Elda M. Melchor-Martínez, Rodrigo Macías-Garbett, Lynette Alvarado-Ramírez, Rafael G. Araújo, Juan Eduardo Sosa-Hernández, Diana Ramírez-Gamboa, Lizeth Parra-Arroyo, Abraham Garza Alvarez, Rosina Paola Benavides Monteverde, Karen Aleida Salazar Cazares, Adriana Reyes-Mayer, Mauricio Yáñez Lino, Hafiz M. N. Iqbal, Roberto Parra-Saldívar

**Affiliations:** 1Tecnologico de Monterrey, School of Engineering and Sciences, Monterrey 64849, Nuevo Leon, Mexico; elda.melchor@tec.mx (E.M.M.-M.); rodrigo.macias@tec.mx (R.M.-G.); a00814259@tec.mx (L.A.-R.); rafael.araujo@tec.mx (R.G.A.); eduardo.sosa@tec.mx (J.E.S.-H.); diana.ramirez.gamboa@tec.mx (D.R.-G.); a01036078@tec.mx (L.P.-A.); 2Cadena Comercial OXXO S.A de C.V., Monterrey 64480, Nuevo Leon, Mexico; abraham.garza@oxxo.com (A.G.A.); rosina.benavides@oxxo.com (R.P.B.M.); karen.salazar@oxxo.com (K.A.S.C.); 3Centro de Caracterización e Investigación en Materiales S.A. de C.V., Jiutepec 62578, Morelos, Mexico; areyes@cecim.com.mx; 4Polymer Solutions & Innovation S.A. de C.V., Jiutepec 62578, Morelos, Mexico; myanezl@polysol.com.mx

**Keywords:** bioplastics, biobased plastics, plastic degradation, bioplastic sources, plastic business case, bioplastic legislation

## Abstract

Plastics have become an essential part of the modern world thanks to their appealing physical and chemical properties as well as their low production cost. The most common type of polymers used for plastic account for 90% of the total production and are made from petroleum-based nonrenewable resources. Concerns over the sustainability of the current production model and the environmental implications of traditional plastics have fueled the demand for greener formulations and alternatives. In the last decade, new plastics manufactured from renewable sources and biological processes have emerged from research and have been established as a commercially viable solution with less adverse effects. Nevertheless, economic and legislative challenges for biobased plastics hinder their widespread implementation. This review summarizes the history of plastics over the last century, including the most relevant bioplastics and production methods, the environmental impact and mitigation of the adverse effects of conventional and emerging plastics, and the regulatory landscape that renewable and recyclable bioplastics face to reach a sustainable future.

## 1. Introduction

Since the creation of the first synthetic polymers in the 20th century, plastics have become an essential material in human activities due to their valuable characteristics, such as their low cost of production and broad potential applications. It is estimated that 370 billion tons of plastic materials are produced each year, a trend that is not expected to diminish in upcoming years [1]. Plastics are highly praised due to the ease of their production and diversity of applications thanks to their properties of durability, sturdiness, and lightweightedness, which are unmatched by other material categories and the current proposed alternatives [2].

Although there are an estimated 60,000 different plastic formulations, only six polymers account for 90% of the total plastic production: polypropylene (PP), high-density polyethylene (HDPE), low-density polyethylene (LDPE), poly(ethylene terephthalate) (PET), polyvinyl chloride (PVC), polystyrene (PS), and polyurethane (PU) [3]. The most common synthetic polymers are manufactured from petroleum-derived compounds, a nonrenewable resource that has raised concerns over its long-term sustainability, as its fabrication model uses declining resources at increasing prices [4]. Moreover, these resources are not being used efficiently. Single-use plastics consume 1.6 billion liters of oil and are disposed of immediately after use, with a limited possibility of recycling [5].

The lack of management strategies for plastics once they reach the end of their lives, coupled with the inherent durability of polymeric materials, has given rise to the improper disposal of plastic products and the accumulation of plastic-related particles. This issue has been exacerbated by single-use plastics and an increase in the production of personal protection equipment brought on by the COVID-19 pandemic [6]. Plastic is the predominant type of human waste found in freshwater reservoirs, accounting for 57% of the total debris by weight [7]. Discarded plastics subjected to natural degradation turn into secondary microplastics, an emerging contaminant category of concern that often appears in modern research due to its potential effects on the environment and wildlife, as well as its status as a vector for adjunct pollutive particles such as heavy metals and hormone disruptors [8].

The critical impact of plastics on the environment as well as on human health, coupled with the need for sustainable methods to produce synthetic plastic analogues to alleviate the global demand, has been considered by policymakers and scientists in recent years. In the UN Global Sustainable Development goals, Goal #14 specifically mentions the need to mitigate the impact of plastic (Indicator 14.1.1 (b)—extraordinary efforts are required to reduce microplastic discharge into marine and freshwater ecosystems [9]), and another 12 goals relate to the alleviation of the problems created by the plastic industry in the human health and environmental spheres [10]. Nevertheless, the plastic alternatives that are currently presented lack widespread adoption. This is mainly due to their perceived inferiority, the lack of large-scale production processes to sustain their demand, and the higher costs associated with their production [11]. In recent years, bioplastics have been proposed as a promising alternative to conventional plastics because of their similar properties. The fabrication of this branch of polymeric materials has certain advantages over petroleum-based resins, such as their renewable nature and lower costs of industrial-grade production [12]. Bioplastics have recently been adopted for a few applications, such as single-use plastics and 3D-printing materials; however, their sustainability and ecocompatibility have remained a topic of discussion. Their widespread usage has been hindered by technical limitations in the production processes, as they offer lower yields than their petroleum-based parallels [13].

This article summarizes the historical relevance of plastic materials, their adoption and evolution, their current shortcomings, and the emerging trends in biobased plastic manufacturing and implementation. The environmental effects of conventional and emerging plastics are approached, as well as their mitigation strategies and the main applications of recent alternative materials. The challenges and opportunities of these biological plastics are discussed, as well as the relevant market in the context of the modern demands for plastic materials. The relevant legislation that concerns plastic production, certification, and the regulation of its disposal is also briefly discussed.

## 2. Historical Development of Plastic Materials: A Brief Timeline

The field of plastics is currently extremely prevalent in research and innovation, particularly with relation to new polymeric products, with a focus on their application and performance. Nevertheless, some concerns remain regarding their production, commercialization, and final disposal. The synergy between biotechnology, bioeconomy, and the chemical industry provides possible solutions to these problems [14].

Records of the use of plastics date from as early as a few centuries ago. Ancient civilizations around the world used materials such as resins, insoluble oils, and amber similarly to how we use plastics today. The first reference to rubber concerned cultures native to Central America who congealed latex to create waterproof shoes. In 1839, Charles Goodyear, an American inventor, discovered the elasticity and resistance of rubber heated with sulfur (Figure 1). This process was termed vulcanization and patented in 1844. Subsequently, ebonite was created and became relevant because it was thermosetting and was prepared from a natural material, rubber, though with larger quantities of sulfur. In the 1850s, the conditions for controlling the nitration of cellulose were optimized in Europe. A solid residue was produced from solvent evaporation, which demonstrated elastic and waterproofing properties. In 1862, Parkesine was prepared by the dissolution of cellulose nitrate in minimal solvent [15]. In 1863, two materials that had similar characteristics to Parkesine were developed: xylonite and ivoride. In 1869, billiard balls were being made out of cloth, ivory dust, and shellac in the US. In 1872, the term celluloid was first used to describe a material obtained from cellulose nitrate and camphor. The first protein-based member of the natural polymer family was developed in Germany in 1897 out of casein that was reacted with formaldehyde. Casein was separated from milk by coagulation. The formalized casein was used in buttons, dress ornaments, necklaces, manicure sets, pens, and other decorations. In 1899, ebonite was patented in the UK. It was made by reacting phenol and aldehyde resins and was used as an electrical insulation. In 1927, a nonflammable replacement for celluloid was created, called cellulose acetate, which was extensively used for artificial fibers [16].

Between 1930 and 1940, the current major industrial thermoplastics derived from ethylene were created: polystyrene (PS); poly(vinyl chloride) (PVC); and polyolefins, such as polyethylene (PE), polypropylene (PP), and polymethylmethacrylate (PMMA). During World War II (WWII), PMMA, a rigid, transparent plastic, was used for aircraft glazing and, to a lesser extent, in denture manufacture. The first polyurethane was synthesized by Otto Bayer in Leverkusen, Germany, in 1937 [17]. The diversity of raw materials that could act as sources for polyurethane production, as well as its wide range of uses, enabled the wide application of plastics in settings such as buildings, automobiles, coatings, and sealants [18].

More materials were designed before World War II. One was nylon, a sticky, bendable material, first formulated in 1933. Afterwards, during WWII, polyamide 66 and Teflon were discovered in 1941. During the mid-1950s, high-density polyethylene (PE) was produced, followed by polypropylene (PP). In 1956, polycarbonates were developed in the United States and Germany simultaneously. A variety of copolymers and blends were produced during the period 1960–2000. In the 1960s, there was a growth in the variety of synthetic fibers. Two important fibers to note were Nomex^®®^ and Kevlar^®®^, the first meta-aramid and para-aramid fibers created by DuPont^™^. Nomex^®®^ had a higher melting temperature [19], while Kevlar^®®.^ had a structure that allowed it to create composites [20], replacing steel fibers in racing tires and later being used for consumer products and human armor [21].

Polybutylene terephthalate was introduced in 1969, and later, polycyclohexylenedimethylene terephthalate, which was a plastic polyester with a higher melting temperature. High-performance thermoplastics were introduced during the 1970s and 1980s and could withstand temperatures of above 200 °C. In 1977, polyetherether ketone (PEEK) was invented, followed by polyether sulfone (PES) in 1983. In 1990, polyhydroxybutyrate was commercially produced under name “Biopol”. In the late 1990s, ethylene, propylene, and styrene were first introduced [22]. In the 2000s, the field of plastics was focused on materials produced from vegetable sources. Considerable research was carried out in bioplastics such as polyhydroxyalkanoate and poly-lactic acid (PLA) extracted from sugarcane, corn, and rice [23]. They were considered alternatives to petroleum-based plastics and could be biodegraded. Biobased polymers such as bio-polyethylene (bio-PE) and bio-poly(ethyleneterephthalate) (bio-PET) have been produced for its functionality and considerable capacity in packaging application. In 2019, biobased polypropylene was produced at a commercial scale, and its production capacity is expected to quadruple by 2025 [12,24].

## 3. Environmental Impact and Health Effects of Synthetic Plastics and Bioplastics

### 3.1. Plastic Degradation and Insertion into the Environment: An Overview

The affordability and desirability of plastic have enabled its ubiquity in all aspects of human development. Plastic production reached 368 million metric tons in 2019 and is projected to reach a total of 1.1 billion metric tons by 2050 [1,25]. Despite recent efforts, local and global policies for plastic disposal and handling after products have reached the end of their lifetimes have lagged behind production and consumption patterns, promoting the unregulated discharge of plastic materials into the ecosystem [26]. The stability conferred by the polymeric nature of plastics hinders their degradation in the environment by natural means. This results in the accumulation of plastics in the ecosystem as well as the macroscopic fragmentation of plastic particles through mechanical erosion, ultraviolet weathering, and biological assimilation, meaning that extended periods are needed for their complete breakdown beyond just their chemical degradation [27].

Once a plastic product is disposed of, its final fate can vary according to its geographical location; the available waste management infrastructure; the economy; and the intrinsic properties of the discarded product, such as its morphology and composition. Plastic waste can either be primed for reuse or recycling, managed through landfilling and incineration, or directly disposed of into the environment [1]. Unlawful means of plastic disposal such as littering and unregulated landfills also represent a significant source of plastic entry into ecosystem elements such as rivers and soil [28]. The plastics’ morphology and chemical nature also determine the polluting potential of discarded products. The diversity and complexity of the interactions between plastic sources, compositions, morphologies, entry pathways, and degradative mechanisms hamper the development of a holistic understanding of the global plastic pollution issue, preventing practical and integral actions to mitigate its environmental impact (Figure 2) [29].

As mentioned previously, the chemical nature of plastic hinders its quick breakdown in the environment. It tends to accumulate in soil, freshwater reservoirs, and oceans. Plastic permanence is estimated to last hundreds or even thousands of years, depending on factors that facilitate passive ageing and degradation. The onset of polymer decay is triggered by ultraviolet degradation and thermo-oxidative reactions promoted by the sun and environmental components such as beaches and pavements [30]. Plastics may also degrade mechanically through erosion caused by continuous contact with rocks, wind, and water, which shear and tear plastic particles [31]. Finally, plastics may also degrade through biological means, either by the chewing and digestion of macroscopic animals or by the microscopic biodegradation mediated by bacteria, fungi, and other biologic actors (Figure 3) [32].

The natural routes of watersheds transfer plastic waste from soil-based landfills into water bodies. Although pollution models have traditionally considered oceans as the final destination of plastic waste, freshwater networks have recently been recognized as both an active transport vector and a retention platform for polymeric waste [33]. The size and composition distribution of plastic pollutants in rivers is heterogeneous and characterized by retention in riverbeds, sediments, and vegetation reported throughout the world in landmarks such as the Great Lakes [34], the Danube River [35], the Ganges River [36], and the Thames River [37], to cite some examples. Polymer particles may then be ingested or absorbed by organisms, inserting plastics into the ecosystem’s trophic chain [38]. The presence of plastic particles in freshwater also poses a threat to human health, as it is the primary source of drinking water worldwide, and the presence of microplastics has been reported in local water-supply networks [39,40]. Oceans represent the major worldwide sink for discarded plastic through natural transport pathways, and this is exacerbated by the indiscriminate product disposal caused by human activities such as fishing, commerce, industry, and tourism. Oceans are thus cited as the ecosystem most affected by human plastic pollution, which impacts the marine biosphere in all degrees and interactions [41]. Marine plastic debris floats and is diffused by global currents, acting as a carrier and a passive modifier of local ecosystems, introducing foreign particles and organisms, and potentially disrupting local environment interactions. Marine plastic is also deposited on beaches and shores. The persistence of marine plastic also favors material inflow as opposed to outflow mechanisms such as degradation, leading to product accumulation, with the Great Pacific Patch being the most well-known marine plastic reservoir in the world [42]. The uptake of plastic particles by living organisms has also been studied and reported at macroscopic as well as microscopic trophic levels; nonetheless, the fate of this uptaken plastic, whether retained in tissues or excreted, is poorly understood [43].

Plastic fragments in soil are accumulated and distributed according to their particle size and the local environment dynamics, becoming a feature that alters soil properties and health. Microplastics can hinder soil density and bulk volume, affecting water retention and promoting the diffusion of other pollutants by creating surfaces with high adsorption coefficients [44]. The presence of plastic particles in terrestrial ecosystems negatively impacts biological developments at macroscopic and microscopic levels. Plastic debris may be ingested by soil-based organisms such as earthworms that act as microplastic concentrators, accumulating in their tissues [45]. Microplastics can thus be found in the terrestrial food supply chain through the ingestion of such organisms by poultry, later leading to negative implications for human health and posing a severe risk of microplastic exposure and ingestion [46]. An investigation conducted by Sun et al. [47] demostrated that polymer particles can also impact soil bacterial consortia, changing the composition of bacterial communities directly related to microplastic concentration and morphology.

The pathways through which plastic and related contaminants enter the atmosphere have been proposed, yet there still remains a knowledge gap. The incineration of plastic products is a common waste-management strategy for energy recovery, along with recycling and landfilling, as these are proposed as low-cost solutions that require minimal space [48]. Nevertheless, plastic incineration leads to the emission of hazardous contaminants and poses a risk to human and animal health while disrupting the environment through the release of carcinogenic and mutagenic compounds [49]. Plastic debris may also enter the atmosphere by mechanisms of natural erosion of predominantly synthetic textiles in the form of microfibres [50]. The fate of airborne plastic particles may vary according to the local ecosystem conditions, and they can be deposited into water ecosystems and soil, where they can be become airborne again in a dynamic cycle [51]. Plastic particles can also be inhaled and deposited in animal respiratory tracts, providing another pathway for trophic accumulation and transport [52].

### 3.2. Macroplastics

Macroplastics constitute a significant component of human litter and are commonly defined as plastic pieces over 25 mm in size [53]. This class of plastic debris impacts landscapes and ecosystem biology and serves as a source of secondary microplastics [54]. It is estimated that macroplastics kill 1 million marine animals through ingestion and entanglement [55], and the recent onset of the COVID-19 pandemic has exacerbated the disposal of single-use plastic products in the form of personal protection equipment, packaging, disposable cutlery, and containers [56,57].

Macroplastics have a negative impact on the environment due to their several hazardous effects. Depending on their nature, discarded plastic products have the potential to entrap organisms, hindering their mobility and visual capacity [58]. Macroplastics also act as vectors for the conveyance of undesirable ecosystem elements and pollutants through agglomeration and passive transport through water and air currents, exacerbated through climatic events such as rain and hurricanes [59], and may introduce nonindigenous species to foreign biospheres, such as mollusk stocks and algae dispersal through a mechanism known as rafting [60].

Animals may actively ingest macroscopic products which can be regurgitated, excreted, or retained for extensive periods [61]. In the latter case, plastic pieces large enough may cause gut blockage, starvation, or reduced nutrient absorption, leading to animal mortality [62]. Over time, plastic ingestion may act as an evolutionary trap, affecting the viability of animal populations and overall species through the persistent risk of prey resemblance and ever-increasing animal–plastic encounters [63].

Sizeable plastic debris is also likely to accumulate at drainage basin constrictions, causing blockages and leading to increased risks of flooding and raised water levels [64]. In urban settings, the blockage of drainage and sewer systems causes undesirable effects such as bad smells in streets and propitiates the spread of diseases such as cholera and typhoid through contaminated drinking water [65]. In natural ecosystems, water obstructions promoted by plastic accumulation may promote changes in the hydrodynamics of rivers and lakes, altering the immediate landscape by the accidental creation of ponds during rainfall, leading to biofouling [66]. Floods caused by plastic blockages also provide a breeding opportunity for vector arthropods, increasing the risk of infectious viral and parasitic diseases in prolonged inundated zones [67].

Macroplastics can have adverse effects on soil, as they disrupt its physical and chemical properties. Plastic particles on land can alter the floor salinization and the root insertion of neighboring plants, with potential effects on the emergence of crops of agricultural interest [68]. Macroscopic plastic debris has also been evidenced to directly block photosynthesis and entangle plant seedlings, having a considerable impact in plant survivability in ecosystems deeply affected by littering [69]. Soil can also preserve plastic through earth-borne biota, such as earthworms, which are reported to burrow plastic particles that hinder solar-dependent degradation through UV radiation and lead to slow plastic decomposition [70].

### 3.3. Microplastics

The term microplastic has been commonly used in literature to describe a broad assortment of plastic-derived particles classified according to their size. Although the term is widespread and standard, the complexity and variety of microplastic compositions and morphologies are often overlooked in categorizing these contaminants [71,72]. This simplification has led to the inadequacy of studies and protocols that assess the environmental effect of microplastics, the irreproducibility of strategies for microplastic detection, and the ineffectiveness of pollutant-mitigation governance strategies [73].

Microplastic particles have become ubiquitous in all environments, and their presence has been reported in air, water, and soil ecosystems, as well as in food [74]. The primary sources of entry into the environment are either the release of synthetically designed plastic microparticles included in functional products such as cosmetics, drugs, and pellets (primary microplastics) or by the natural degradation of larger plastic debris through the previously discussed processes (secondary microplastics) [75]. The composition of the plastic in addition to its physical properties, environmental effects, and exposure time define a microplastic particle [30,76]. Its interaction with environmental elements, accumulation, and association with other plastic and pollutive particles, given by its physical properties, define its effects on the ecosystem [77]. Microplastic particles have a variety of shapes, including spheres, fibers, fragments, and films, each related to their source and composition with a varying pollutive potential through their macroscopic and microscopic biological effects [78].

The emission of plastic residues into aquatic environments is estimated to be between 4.8 and 12.7 million metric tons annually [79]. A fraction of such debris is degraded to create microparticles, along with the direct discarding of primary microplastics through water drainages. Once in oceans and freshwater reservoirs, microplastic can be ingested by fish and plankton and be inserted into food chains through trophic exchange into larger predators and humans, ultimately affecting these species through the consumption of contaminated prey [52]. Microplastics can float and drift in the water surface of oceans and rivers, interfering with surface microbiota and inhibiting the photosynthetic efficiency of algal plants, diminishing available nutrients in the marine ecosystem [80]. Degraded microplastic particles can also be deposited in the marine environment. Once there, they act as pollutant liberators, slowly releasing adjunct chemical pollutants such as endocrinal disruptors (BPA), while their further degradation is limited due to the inhibition of ultraviolet degradation that converts river and ocean beds to central microplastic repositories [81,82].

Microplastics possess a large surface area and adsorptive potential, which allow them to interact with other organic molecules, heavy metals, and other microplastics, making them a pollutant vector [83]. In the case of heavy metals, PVC and PS microparticles have been reported to have a solid interaction with copper and zinc, common elements found in chemical additives such as paint [84,85]. Bioactive molecules such as pesticides and antibiotics have also been reported to have sorption activity through microplastic molecules, making them a bigger hazard in estuaries compared to oceans and freshwater due to a higher concentration of contaminant particles [86].

Human life is threatened by microplastics through several means. The ingestion of plastic particles through contaminated food is considered the main route of human exposure and can have molecular effects in cancer, obesity, and other oxidative-stress-related conditions. Additives found in microplastics, such as bisphenol A, are also endocrine disruptors that hinder proper development and have adverse reproductive effects [87]. Microplastic particles can also be inhaled, accumulating in lung tissue and causing cytotoxic effects in pulmonary cells, resulting in bronchial inflammation, fibrosis, allergic reactions, and interalveolar lesions [88]. Human exposure to airborne microplastics is estimated at 272 particles per day and is influenced by the material’s nature, the performed activity, the quality of ventilation, and the particular season [89]. It is estimated that the primary microplastic morphology in the atmosphere is microfibers composed of nylon, polyester, and acrylic [90].

### 3.4. Other Plastic-Derived Pollutants

Besides the polymeric backbone, other chemicals are frequently added to synthetic plastics to heighten the qualities of the final product, such as the color, mechanical features, and stability. The classification of additives is as diverse as that of plastic polymers and includes plasticizers, antioxidants, dyes, lubricants, and fillers [87]. The release of chemical pollutants in conjunction with plastic debris particles may occur at all steps of the plastic lifecycle. The release of such compounds is mediated by the composition of the plastic and additive blend, the dynamic of the plastic particle with the additive, and the interaction of the compound with the environmental features (e.g., leachate solubility) [91].

Plasticizers are frequently employed to improve polymeric mechanical features, such as the flexibility and processability of polymer resins [84]. Over 90% of produced plasticizers are used to enhance PVC properties, expected to reach 59 million tons in 2020 [92,93]. Phthalic acid esters are the most used plasticizers globally, di(2-ethylhexyl) phthalate (DEHP) being the most pervasive plasticizer added to plastic [94]. Phthalates are not covalently associated with the polymeric matrix, and the mechanical wearing of the plastic product allows its migration to food and the environment [93]. Phthalate esters are recognized as a predominant category with contaminant effects, and their occurrence in all aspects of the environment has been comprehensively reported. This category of plastic additives has been recognized as potentially hazardous to the environment and human health through endocrine disruption [95]. DEHP also has a low toxicity threshold (LC_50_ of 0.50 ppm), causing embryonic mortality and necrosis in potentially exposed animals such as zebrafish [96,97].

The deleterious effect of phthalates on human health is documented and includes the alteration of estrogenic levels, infertility, asthma, diabetes, hyperglycemia, and endometriosis [98]. Phthalate exposure has also been associated with an increased BMI and the diagnosis of obesity in children [99]. The effects of DEHP exposure may also increase multigenerational hepatic fibrosis risk through the alteration of regular DNA methylation patterns [100]. Microplastics are considered a source of phthalate emission in the environment, and the high hydrophobicity and surface area of plastic particles enable the adsorption and transport of phthalates into water, soil, and sediment deposition [101]. The most frequent human entry pathway for such a family of compounds is ingesting contaminated foods that have come into contact with phthalate-supplemented packaging [102], followed by microplastic consumption through the drinking of contaminated water sources [103].

Different kinds of technologies can also promote the oxidative degradation of synthetic plastic based on fossil resources. This approach is frequently proposed as a pathway to minimize the environmental impact through the enhanced breakup of plastic resins without compromising their functionality [104]. Pro-oxidative additives are based on the salts of transition metal ions such as Co^2+^, Fe^2+^, or Ni^2+^, which stimulate the degradation of polymeric chains in the presence of oxygen and heat or radiation [105]. The most widely used commercial oxodegradable additives are d2w and TDPAs (totally degradable plastic additives); both are based on a combination of manganese, iron, cobalt, and nickel salts [106]. Oxodegradable plastics have enjoyed widespread adoption recently due to the technical feasibility of their implementation in already established manufacturing processes [107].

Concern has been raised over using transition metal compounds as additives designed to degrade environmental elements. Oxoplastics may then contribute and accelerate the production and accumulation of microplastics worldwide, and legislative regulations have recently been imposed due to the ambiguous nature of the conditions needed to degrade the modified resins completely [108]. The incomplete fragmentation of oxodegradable plastics also challenges their proenvironmental viability, as studies conflict with the true degradative capabilities of discarded oxoplastics. The usage of pro-oxidative additives does not ensure the complete degradation of plastics and may require special conditions nonreplicable in the environment [109]. However, transition metal salts have been regarded as safe due to their design as fatty acid salts of potential nutrient microelements [110].

Research has argued that the degradation pathway of oxoplastics may be further assisted by biological mechanisms. The term oxo-biodegradable plastics refers to the two-stage degradation of additive-enhanced polymers. The first stage of degradation occurs by physical phenomena such as ultraviolet light and erosion, and the resulting plastic particles are then degraded by microorganisms through enzymatic pathways [111]. The addition of an oxodegradable additive has been found to enhance complete biodegradation with no toxicological effects on the obtained compost [107]. The resulting oligomers from photo-oxidized plastics have also been used as a carbon source for bacterial β-oxidation. Such a phenonmenon was observed when Rhodoccocus rhodochrous was observed to assimilate polyethylene film oligomers, leading to their biodegradation [112]. Both degradation mechanisms are considered biologically compatible, and the degradation of oxoenabled waste can be accelerated by metabolic action through compatible enzymes [113]. The findings of such reports show that the use of oxidant additives to promote plastic degradation may be justified from a combined physical and biological mechanism standpoint.

The Parliament of the European Union has carried out a ban on oxodegradable plastics under the “lack of evidence that oxo-degradable plastics are fully degraded in a reasonable time, are not appropriate for recycling or composting, and there is a risk that small plastic pieces will not completely biodegrade” [104]. Bioplastics are a better alternative in terms of complete product degradation versus oxo-bioplastics [114].

## 4. New Generations of Plastics

Bioplastics are polymers that can be biodegradable or not. Additionally, they can be synthesized from renewable biological sources such as bacterial, plant, and algal sources. However, their biodegradability depends on the chemical structure, but not on the sources used [115,116]. Nowadays, bioplastics represent only around one percent of the total plastic produced annually. However, it is expected that the demand for bioplastics will increase [117]. Figure 4 shows the current stages of bioplastic production, from the research and development stages to the commercial stage.

Bioplastics are generally reported in three categories (Figure 5): (i) fossil-resource-derived but biodegradable, (ii) partially biobased or biobased and non-biodegradable, and (iii) biobased and biodegradable [118,119].

### 4.1. Biodegradable Plastics from Fossil Resources

Generally, synthetic polymers are obtained from crude oil or natural gas. Most of these polymers are not biodegradable. Nevertheless, degradability can be achieved by integrating unstable bonds (e.g., amide, ester, or ether) [120]. Examples of biodegradable fossil-based plastics are poly(1,4-butylene adipate-co-1,4-butylene terephthalate) (PBAT) and polycaprolactone (PCL). PBAT is a polyester that is synthesized by polycondensation from the combination of dicarboxylic acids and diols. PBAT, due to its aliphatic unit in its molecule chain, has good biodegradability and excellent mechanical properties [121]. PCL is a hydrophobic and biodegradable polymer with sufficient mechanical strength and flexibility [122]. It has excellent biocompatible properties that enable its use for medical applications. To enhance their properties, PCL and PBAT can be used in blends with biobased and biodegradable materials such as polylactic acid (PLA) and polyhydroxyalkanoates (PHAs), and polybutylene succinate (PBS) [120].

Oxo-biodegradable plastics are composed of petroleum-based polymers such as polyethylene (PE) with pro-oxidant additives that promote the degradation process [14]. The materials used as additives are generally transition metals such as iron, manganese, cobalt, and nickel. The additive’s function is to break down the large molecular polymer chain into smaller fragments that microorganisms can process and convert into biomass and carbon dioxide (CO_2_) [104,123]. The degradation of these plastics can be initiated by UV light, moisture, heat, and microorganisms. Usually, the degradation can be evaluated by measuring the changes in physical properties such as the loss of molecular weight, the amount of CO_2_ evolved, and the microbial growth on the polymer surface. The degradation of oxo-biodegradable plastics generally takes months to years; however, this length of time is occasionally unpredictable, because it depends on climate factors such as temperature and the intensity of solar radiation. The most commercially successful additives used as pro-oxidants are totally degradable plastic additives (TDPAs), Renatura, AddiFlex, d2W, and Reverte. AddiFlex is employed to produce single-use plastic bags. Reverte is used for bottle production, and different companies use d2W for their consumables, such as Pizza Hut, Walmart, and KFC [104].

### 4.2. Biobased Non-Biodegradable Plastics

Biobased plastics that are not degradable represent a group of biopolymers such as bio-polyethylene (bio-PE), bio-poly(ethylene terephthalate) (bio-PET), bio-polyamides (bio-PAs), and bio-polypropylene (bio-PP) [115]. They can be synthesized from biobased sources and offer a nearly equal chemical structure and properties to their fossil equivalents. Among them, bio-PET is the most widely produced bioplastic [124]. Since 2010, the precursors of PET, ethylene glycol and terephthalic acid (TPA), have been obtained from biological sources. Nowadays, the Toyota Tsusho Corporation, Japan and Futura Polyesters, the Coca-Cola–Gevo Venture, and the PepsiCo–Virent Venture are the principal producers of bio-PET [115]. Bio-PE is obtained from biological resources by the dehydration of bioethanol, obtained from glucose. Different natural feedstocks can be used to obtain glucose, such as maize, wheat, sugar cane, and sugar beet. The biopolymer is identical in its chemical, mechanical, and physical properties to fossil-based PE [115]. According to European Bioplastics [117], biobased but non-biodegradable plastics represent almost 40% of the global bioplastic production capacity; nevertheless, it is predicted that in the following years their participation will be lower.

### 4.3. Biobased and Biodegradable Plastics

#### 4.3.1. Starch

Starch is an essential polysaccharide that plants synthesize and store in their structure as an energy reserve. It is considered one of the best biopolymers with extraordinary potential, because it is biodegradable, renewable, and available in huge quantities at a low cost. Starch could be extracted from different sources such as corn, wheat, potato, rice, tapioca, tam, and barley. However, most starch is produced from maize [104,125]. Starch polymers are divided into two principal forms, amylose and amylopectin; the composition of these components can affect the properties of starch-based films [126]. Native starches have limits in their mechanical properties, thermal stability, and brittleness; that is why plasticizers (sorbitol, glycol, and glycerol) are necessary to improve the bioplastic functionality, transforming starch into thermoplastic starch [104]. Additionally, to enhance starch’s characteristics, attempts have been made to mix it with synthetic polymers, lignocellulosic biomass, or agricultural waste (Table 1) [125]. Different studies demonstrate the applicability of starch bioplastics in shopping bags, food packaging, agriculture, and medicine.

#### 4.3.2. Cellulose

Cellulose is a polymer which is renewable and is present in large quantities. It is widely used because it is inexpensive and biodegradable. Cellulose is a polysaccharide chain that contains D-glucopyranose units joined by β-1,4-glycoside linkages. Cellulose from plants can be found from wood and non-wood plant lignocellulosic biomasses. It is the main constituent of plant fiber [139]. Cellulose is a potential film material with a highly crystalline structure; however, it is not soluble in water or common organic solvents. For this reason, it needs to be modified to transform it into water-soluble materials [139]. Cellulose acetate is a derivative of cellulose obtained from a chemical modification of cellulose. It is extensively used in the production of membranes, cigarettes, and food packaging [138].

Although cellulose is a plant material, some bacteria can produce cellulose, which is known as bacterial cellulose (BC) [139]. This is an interesting material due to its unique properties, making it an ideal candidate for industrial-scale production [140]. It has excellent mechanical strength and degradability. Additionally, it is purer than natural cellulose, and it has a higher water-holding capacity. These characteristics make it a promising natural polymer with multiple applications in medicine, food, electronics, and other fields [141]. Few companies produce this polymer on a large scale, such as Bowil Biotech (Poland) [140]. However, the high-cost production has limited its applications. Bacterial cellulose can be synthesized by multiple bacteria species, such as *Acetobacter* sp. and *Gluconacetobacter* sp. [141]. Using by-products and wastewater as raw materials for the synthesis of biopolymer is a strategy for large-scale production at a low cost (Table 2) [142]. The production of BC from agricultural and industrial waste uses food wastes [141,143,144], wastewater, and crude effluents [142,145,146,147]. An important advantage of using these residues is the reduction in both the cost of production and human waste [146].

#### 4.3.3. Polybutylene Succinate

PBS is an aliphatic polyester that is synthesized by the condensation of succinic acid and 1,4-butanediol. Traditionally, the monomers were synthesized from fossil resources; however, butanediol can be prepared from renewable sources such as renewable biomass resources and succinic acid from sugar fermentation (Table 2) [148]. Furthermore, PBS has good mechanical properties comparable to PE and PP, resulting in multiple applications for food packaging, shopping bags, agriculture mulch film, and hygiene products [120]. Nevertheless, the production cost is high, so future developments are needed to make it more economically viable [51].

#### 4.3.4. Polylactic Acid

Polylactic acid (PLA) is an aliphatic polyester produced by chemical or biological methods from cellulosic biomass [120]. It is biodegradable, and its mechanical properties are equivalent to those of petroleum-based plastics such as PET and PS, in addition to properties such as biocompatibility, innocuity, and compostability. The Food and Drug Administration (FDA) allowed the direct contact of the material with food. PLA is produced from lactic acid through the fermentation of renewable resources such as rice, corn, and potatoes. The fermentation is performed principally by lactic acid bacteria; however, different microorganisms have also been used (Table 2) [104]. PLA can be composed of D- or L-lactic acid isoforms or by both. It is necessary to obtain both isomers in separate processes [149], as the crystallinity and thermal stability of the final product depends on the isomer proportion.

PLA is probably the most well-known bioplastic and can be synthesized from renewable and natural resources. Its high manufacturing cost must be reduced in order to produce more bioplastics. Several works in the literature have reported the use of lignocellulosic wastes and food-derived wastes, such as spent coffee grounds [150], fruit wastes [149,151,152], alfalfa silages [153], and corn cobs [154] (Table 2). Despite PLA being extensively adopted, it still has drawbacks such as its cost and its structural breakability and fragility. Different approaches have been studied to improve its properties, such as the incorporation of nanoparticles [155].

#### 4.3.5. Polyhydroxyalkanoates

PHAs are natural aliphatic polyesters that consist of R-3-hydroxyalkanoate groups. There are synthesized by fungal and bacterial strains [156] from different substrates such as industrial by-products, oils, fats, lignocellulosic raw materials, agroindustrial waste materials, sugars, and wastewater (Table 2) [157]. PHAs are less porous and therefore trap less O_2_, CO_2_, and H_2_O, making them suitable for the production of packing materials such as films, coatings, bottles, and bags [156]. Additionally, PHA has good thermal-mechanical characteristics similar to synthetic polymers; therefore, it can substitute for PE and PP [118]. PHBs are one of the most studied polyhydroxybutyrates [158]. PHAs are commercialized by different companies, for example, Metaboli (Woburn, MA, USA); Procter & Gamble Co., Ltd. (Cincinnati, OH, USA); Tianjin Green Bioscience Co., Ltd. (Tianjin, China); Bio-on (Emilia Romagna, Italy); Biocycle PHB Industrial S.A. (Serrano, SP, Brazil); and Goodfellow Cambridge, Ltd. (Huntingdon, UK). The commercialization of this biopolymer has been increased due to the progress in purification technologies. However, in 2020, PHA biopolymers represented only 1.70% of the worldwide production of bioplastics [117]; the high production costs represent the principal reason for the limited production. To make the production of PHA competitive, it is necessary to use low-value waste materials as a substrate [159]. Different waste streams have been evaluated (Table 2), such as waste vegetable oil [160,161] and fruit waste [159,162,163].

**Table 2 polymers-14-01203-t002:** Biopolymer production by microorganisms using agroindustrial residues as a substrate.

Biopolymer	Microorganism	Production Scale	Employed Substrate	Productivity	Reference
Bacterial cellulose	*Gluconacetobacter xylinum BC-11*		Wastewater	1.77 g/L	[147]
	*Gluconacetobacter xylinus*		Wastewater	0.659 g/L	[145]
	*Komagataeibacter saccharivorans*	Static production in flasks	Crude distillery effluent	1.24 g/L	[142]
	*Gluconacetobacter oboediens*	1 L	Crude distillery effluent	0.85 g/100 mL	[164]
	*Gluconacetobacter sucrofermentans B-11267*	Flask	Whey	5.45 g/L	[143]
	*Gluconaceter xylinus BNKC19*		Pineapple peel	12.3 g/L	[141]
	*Gluconacetobacter xylinum CGMCC No.2955*		Wastewater of candied jujube-processing industry	2.25 g/L	[146]
	*Bacillus cabrialesii*		Grass straw, grass husk, wheat husk, and corn cobs		[144]
PHA ^1^	*Pseudomonas putida KT2440*	^4^ BB (3 L)	Waste vegetable oil	1.91 g/L	[160]
	*Pseudomonas chlororaphis 555*	Pulse-fed batch fermentation (5 L)	Waste cooking oil	13.87 g/L	[161]
	*Pseudomonas resinovorans*	^4^ BB (15 L)	Grease-trap waste	0.41 g/g maximum mcl-PHA^2^ 61.8%	[165]
	*Pseudomonas chlororaphis* subsp. *Aurantiaca*	^4^ BB (2 L)	Diluted fruit pulp waste	0.15 g/g maximum mcl-PHA^2^ 49%	[166]
	*Halomonas campisalis MCM B-1027*	^5^ SF 250 mL	Banana and orange peel	0.329 g/L (banana) 0.11 g/L (orange)	[162]
PHB ^3^	*Bacillus cereus*	^5^ SF 250 mL	Grape peel	0.53 g/L	[159]
	*Bacillus subtilis*	^5^ SF	Papaya and orange peels	11.65 g/L (papaya) 9.68 g/L (orange)	[167]
	*Klebsiella pneumoniae*	^5^ SF 125 mL	Watermelon, papaya, orange, and banana peels	22.61 g/L 23.72 g/L 23.38 g/L 25.11 g/L	[163]
PHB ^3^ and mcl-PHA ^2^	*Cupriavidus necator, Pseudomonas citronellolis*	^4^ BB (10 L)	Apple pulp waste	3.03 g/L	[168]
L-lactic	*Bacillus coagulans*		Sugarcane bagasse	1.7 g/L·h	[152]
	*Bacillus coagulans*		Corn cob residue	79 g/L	[169]
	*Enteroccus mundtii*	350 mL flask	Spent sulfite liquor	56.3 g/L	[170]
Lactic acid	*Bacillus subtillis and Lactobacillus buchneri*		Alfalfa silage	44.2 g/L	[153]
	*Lactobacillus bulgaricus, Strepto- coccus thermophilus, Lactobacillus acidophilus, Lactobacillus plantarum, and Lactobacillus casei.*	1000 mL bottles	Swine manure with apple waste	28 g/L	[151]
	*Lactobacillus rhamnosus B103*		Dairy industry waste	143.7 g/L	[171]
D-lactic acid	*Lactobacillus delbrueckii* ssp. *delbrueckii CECT286*	^4^ BB (1 L)	Orange peel wastes	6.72 g/L·h	[149]
	*Saccharomyces cerevisiae*		Spent coffee grounds	13.4 g/L	[150]

^1^ PHA—polyhydroxyalkanoates; ^2^ bmcl-PHA—medium-chain-length polyhydroxyalkanoate; ^3^ PHB—polyhydroxybutyrates; ^4^ BB—batch bioreactor; ^5^ SF—shake flask.

#### 4.3.6. Other Natural Sources

Seaweed is an excellent alternative for bioplastic production because seaweeds can grow fast and are easy to harvest and cheap. To improve their properties, they can be mixed with other species or materials [172]. Additionally, the use of seaweeds for bioplastics can reduce the impact on the food chain [123]. Standard methods include washing, milling, drying, alkalinization, acidification, neutralization, filtration, and precipitation to produce seaweed bioplastics. Nevertheless, the traditional methods are expensive and have a low yield. For these reasons, green production methods have gained attention due to their high potential and viability [172]. Seaweed polysaccharides can be used for food industry applications, packaging materials, and coatings [123]. Microalgae also represent a promising feedstock for bioplastic production due to their fast growth rates. Numerous researchers have implemented the use of whole microalgae cells such as *Spirulina* sp. [135], *Chlamydomonas* sp. [135], and *Scenedesmus* sp. [173].

Similarly, fungal mycelia from seed peels and corn stalks were used to produce biodegradable packaging and tilling; the mycelia were composed of polysaccharides, chitin, proteins, and lipids. A New York company named Evocative and the Swedish company IKEA have used this bioplastic [123,174]. Chitin and cellulose from crab shells and fiber trees, respectively, have been used to contain liquids and foods [175].

## 5. Green Industry of Plastics

### 5.1. Global Market, Business Cases, and Applications

Currently, plastic products are the material of choice for different industrial, medical, and personal applications, among others. Most plastic products are derived from petrochemicals and are single-use, representing losses of 95% of the material value, which is reflected in annual economic losses between USD 80 and 120 billion [117]. To ensure greater environmental sustainability, it is necessary to exchange non-biodegradable plastics of petrochemical origin for biodegradable and compostable biobased materials. Bioplastics are attractive alternatives due to their rapid biodegradation in the environment into CO_2_ and H_2_O, causing fewer negative effects on the environment [176]. Environmental awareness about the impact of the use of plastics of petrochemical origin has increased both at the business and societal level; however, for economic reasons, bioplastics only represent about 1% of the 360 million tons that are produced annually. The advancement of knowledge and the development or improvement of biopolymers allows the reduction in production costs, and in turn the increased demand for bioplastics is generating an annual growth of 20–30% [177].

Currently, the production of bioplastics uses mainly carbohydrates from plants such as sugar cane or corn as a raw material. Nevertheless, the use of these crops, called primary crops, for bioplastics is in competition with the safety and economic stability of the food industry. Another strategy to produce bioplastics is the use of non-food raw materials, known as second-generation, such as cellulose from crop residues such as corn and wheat stubble, and third-generation raw materials such as micro- and macroalgae. The current production of bioplastics, 2.11 million tons, translates into approximately 0.7 million hectares of arable land, which represents 0.02% of the world’s agricultural area. However, the growing demand for bioplastics will require a greater cultivation area, which will generate greater food competitiveness and is why it is important to implement the sustainable production of bioplastics through effective biotechnological processes from food waste, non-food crops, and cellulosic or algal biomass [117].

The production of bioplastics began due to the need to replace the non-biodegradable plastic materials of single-use products, such as bags, plates, bottles, plastic films, packaging, containers, and cutlery. However, only in recent years has this transition increased due to the decrease in the production costs of biodegradable biopolymers derived from new technological advances and new renewable sources for the production of biopolymers, in parallel with public policies and social conscientization, which have made it possible to produce economically viable bioplastics. Table 3 summarizes the properties of biopolymers in a relation to their applications. Biopolymers on their own do not provide the properties demanded by the market. The costs per kilogram were estimated according to the literature [176].

Throughout the world, various companies have been established that develop and produce biodegradable biopolymers/bioplastics for different applications, which in turn has created greater competitiveness in the bioplastics market to reduce the costs of biodegradable biopolymers (Table 4). The most widely produced biopolymers are mainly PLA, PHA, and starch, which have been used in various applications from packaging to bioplastic materials for toys, parts for the automotive industry, construction, electronics, and agriculture, among others.

The Qmilk^®®^ company has developed one of the most innovative biopolymers, managing to produce a 100% natural polymer from whey with antimicrobial, biodegradable, and compostable properties; it is also a flame-retardant with a low density and high hydrophobicity. This product is created by casein and forms an excellent barrier to gases such as oxygen and CO_2_ and aromas. All these properties give it a great advantage against other biopolymers for food and grocery packaging and textile-fiber production with silklike characteristics [179].

Biofase^®®^ is a Mexican company that has also stood out for the development of biobased biopolymers with avocado seed starch. This company uses an agroindustrial waste widely produced in Mexico to manufacture single-use bioplastics for applications such as cutlery, straws, and food containers [180].

Ingeo^®®^ is a biopolymer developed by NatureWorks^®®^ based on environmental sustainability and a circular economy, using greenhouse gases such as CO_2_ as a raw material to generate a polymer that is biodegradable and compostable. Ingeo^®®^ polymers are currently obtained from sugars produced by plants such as cassava, sugar cane, corn, and beets, which function as CO_2_-sequestering plants, making them sustainable. These polymers are produced by a chain of technological bioprocesses that convert sugars into PLA and are used to create different products such as coffee capsules and electronic components [181].

The development and production of different polymers from biological origins with biodegradable and compostable characteristics have been carried out worldwide, but with greater impact and production in developed countries such as the US and European and Asian nations. Mexico is an example of a developing country where there are still a few limitations to the use of petrochemical plastics. However, the change of some single-use products (bags and packaging) to biobased or oxo-biodegradable products that have a lesser impact on the environment is the way towards a sustainable use of plastics.

Advances in the production chains of biopolymers and the environmental and social commitments of companies have been the result of public policies that limit the use of fossil-derived plastics to encourage a switch to the new generation of plastics. However, with these great changes come new challenges and a need to enhance the production of biopolymers.

### 5.2. Strategies for Plastic Reinsertion and Environmental Impact Mitigation

The recycling and reuse of plastic waste creates new challenges for strategies of collecting, separating, and treating materials of different chemical natures in mixed-waste streams [182]. The diversity of discarded plastic materials and the loss of mechanical properties in recycled polymers have given raise to alternative treatment strategies, such as chemical recycling, which turns plastic waste into chemical products of high value [183]. Biological routes of plastic waste degradation allow its reinsertion into natural carbon cycles, mitigating its environmental impact [184]. Current trends in modern plastic material design, such as biobased plastics, integrate recycling and degradation pathways in the polymer composition [185]. Recent chemical and biological strategies for assimilation are discussed in Table 5.

## 6. Challenges and Opportunities for Biodegradable Plastics from Production to Degradation, and Further Perspectives under Circular Economy

The world demand for plastics has grown rapidly in recent years due to their economic advantages, properties, and applications, with the packaging sector representing about 40% of the demand. As a result of this demand, the production of plastics is growing, while recycling systems are limited [198]. For a long time, there was no need to recover used plastic, since there was not a large demand for these raw materials, the costs were very low, and the impact of the presence of plastics in the environment had not been determined or recognized [199]. 

In recent years, challenges have arisen regarding the scale of production, the cost of production, industrial implementation, recycling, and the effects of degradation and contaminants derived from plastics, as well as issues relating to their regulation. The scale of the production of biodegradable plastics is small compared to plastics from fossil sources. The processes for the production of plastics from agroindustrial wastes such as corn crops and sugarcane bagasse are limited due to issues relating to the logistics of collecting, drying, and storing the raw materials. For example, in PLA production, optimization requires the selection of a microorganism that maximizes the yield [200]. Additionally, the microbial fermentation of lactic acid is associated with high costs derived from the pretreatment of the agroindustrial wastes and their conversion into fermentable sugars for the downstream biotechnological process. The optimization of cost–performance sustainability for bioplastics produced from organic wastes is needed before there can feasibly be a switch from fossil-based plastics to bioplastics [201].

The issues in the industrial implementation of biodegradables plastics are very clear in the food sector. For food packaging, biodegradable plastics should be a barrier against the transfer of gases and water vapor, preferably be transparent for the visual attraction of the customers, and form films with proper tensile and mechanical strength. Currently, the majority of these plastics are opaque, with limited tensile properties and a susceptibility to UV-triggered oxidation [202].

Biodegradable plastics are more susceptible to hydrolysis than petroleum-based plastics, depending on their composition. In the process of degradation, they exhibit a reduction in their molecular weight, leading to a low recycling quality. Their physical and chemical characteristics limit the possibility of mechanically recycling conventional and biobased plastics. The technologies and equipment used and the specific plastic waste influences the recyclability [120].

Additionally, the contamination of plastic waste with organic materials leads to the formation of thin films that often limit the recyclability of petro- and biobased plastics [203]. Of all the plastic produced, only 7% is recycled, about 8% is incinerated, and the rest reaches landfills or the environment [123]. Recyclability is a key element of the circular economy and is of great importance in biodegradable materials such as bioplastics. In the case of plastics of petrochemical origin, the recycling process is already well-established, while for bioplastics, it is still under development, since a fully effective and viable process has not yet been achieved [204]. Most bioplastics of biological origin (PLA, PHA, starch, etc.) may require separation based on their chemical composition; otherwise, they may suffer a loss in their quality and physical integrity due to contamination with other polymers. Currently, the most effective process for the treatment of bioplastics of biological origin is biodegradation [205]. The recycling of bioplastics for food-grade applications is a process that requires a higher degree of processing for the removal of contaminants, which is sometimes not economically viable when compared to the production of enriched compost, which is a more profitable process [176]. Biobased plastics, such as bio-PE, can be recycled by the same processes as their fossil-fuel counterparts, as well as with other plastics of petrochemical origin that are structurally similar [206]. 

The main concerns for the use of petrochemical plastic materials are their low rate of decomposition in the environment, as well as the release of microplastics and various toxic organic compounds that are known to be harmful to the environment and human health. Bioplastics, on the other hand, have a low environmental impact, since their biodegradation does not lead to the release of highly toxic compounds [207]. Different biopolymers obtained from natural resources are easily converted into bioplastics with biodegradable and compostable properties; however, these new bioplastics need regulations that verify and certify these properties to guarantee their safety of use and the lack of harm they pose to the environment [208]. On the other hand, the oxidation of biopolymers during burning releases the same amount of CO_2_ into the atmosphere as petroleum-based polymers. In this sense, strategies to control plastic use and its release are still needed to solve this environmental problem.

Despite the many efforts made in the design of standards to evaluate the compostability and biodegradability of bioplastics, at the industrial level, they have still not been extensively explored and remain unregulated [209].

From the perspective of a circular economy, which aims to properly use renewable natural resources without affecting the environment, biobased bioplastics are the most studied plastic materials, since they present the most advantages in their disposal after use [176]. Adopting a circular economy incorporating biobased resources hopes to reduce greenhouse-effect emissions while increasing the ecoefficiency of resources, valorizing waste and by-products, and thus reducing dependence on nonrenewable resources to reduce the impact of climate change and carbon footprints [210].

The European Commission (EC) implemented an action plan for the circular economy of plastics in 2015. The EC published two legal documents for the packaging industry and its waste management: Directive (EC) 2018/851 of the European Parliament and of the Council of May 30, 2018 which modifies Directive 2008/98/EC on waste, and Directive (EC) 2018/852 of the European Parliament and of the Council of 30 May 2018, which modifies Directive 94/62/EC on packaging and packaging waste. These regulations, based on a circular economy, focused on the reuse and recovery of plastic containers to allow a longer life for the products while minimizing waste and reducing the environmental impact; however, the full recycling potential of the European Union remains untapped, since only 30% of plastics are recycled, with hopes to reach 55% by 2025 [211]. The goal of the “Plastics 2030 Voluntary Commitment” is to recycle 60% of all plastic used and reduce the production of petroleum-based plastic by 100% by 2040 [212]. 

Biorefineries focus on the comprehensive use of different organic wastes such as food, agroindustrial waste, forestry, and algae biomass, which are of great interest for generating new precursors of bioplastics. Through different strategies, such as bioprocesses with microorganisms, the reduction in the use of nonrenewable products has been notable. Different natural resources can be used or explored to obtain biopolymers for bioplastic fabrication, such as PLA, PHA, and PBS. These approaches advance society towards a circular economy and produce different single-use bioplastic products that are transformed into natural compostable compounds without any negative effects on the environment [210].

Biodegradable plastics are not the only solution to the problem of garbage accumulation in landfills and the environment. However, the use of biodegradable bioplastics that are degraded by biological agents under certain conditions and within certain time-frames allows a reduction in their environmental impact, which will leave a significant ecological footprint in the future [114].

## 7. Legislation and Certifications for Biodegradable Plastics

Due to the evolution of bioplastics over the last few decades, their regulation has become imperative. Experts from different fields have been working with organisms across the world to provide standards to analyze the properties, composition, and composability of different bioplastics using specific techniques and conditions that can be replicated. The major organizations creating these standards are the International Organization for Standardization (ISO), the European Committee for Standardization (CEN), and the American Society for Testing and Materials (ASTM). Table 6 is a compilation of the most used and cited standards from different organizations regarding plastics. The United States of America and the European Union currently have many regulations in place, and these are regarded with significant concern, since they have a great impact at the international level. In comparison, in Mexico, there are few reports of standards for bioplastics; meanwhile, the key standardization bodies have a wide variety of standards covering most of the bioplastic family known to this date.

Even when standards are in place, they are not mandatory but are a set of rules and tests to analyze the bioplastic. The portion of the industry sector that analyzes their bioplastic is motivated to obtain a certification for customer approval, particularly when their customers prefer brands with sustainable commitments [213]. The process to obtain the certification is to test the product by an independent certification laboratory that determines whether it complies with the standards and is deserving of a certification. Multiple organizations and associations offer these services and guide the business to obtain certifications that are recognized by governments and society in general. The Biodegradable Product Institute, Tüv Austria (formerly Vinçotte), DIN Certco, the Australasian Bioplastics Association, and the Japan BioPlastics Association are the most in-demand organizations that offer this service (Figure 6). For instance, the Tüv Austria group offers diverse certification logos, which include the Seedling logo, OK compost INDUSTRIAL, and OK compost HOME, all of which are related to the compostability of organic material; the first two logos are acquired when products are in compliance with the standard EN 13432, and the later was developed by the Tüv Austria group without referring to a specific standard, but it contains all the technical requirements that a product must meet to serve as a basis for the creation of other standards across Europe and Australia [214]. Tüv Austria also offers logos that describe the biodegradation of plastic materials depending on the environment in which this process occurs, the logos being OK biodegradable MARINE, OK biodegradable SOIL, and OK biodegradable WATER. All of these are based on the idea of stopping littering and giving information as to the right environment for the biodegradation of a particular material to occur [215]. OK biobased is another logo that rewards companies that manufacture their products with an alternative to fossil-based raw materials. This certification presents the biobased percentage of renewable raw materials used as a base in the product, and the product can be rated up to four stars [216]. Lastly, the logo NEN BIO-BASED CONTENT refers to the biomass content in materials and products in general; this certification is based on the European standard EN 16785-1 and was recently added to the list of the TÜV AUSTRIA group [217].

Notwithstanding the tendency to use bioplastics for their wide-ranging properties, functionalities, and applications, the scheme of regulations and certifications is changing towards more sustainable approaches. Alliances to join the governmental, industrial, and academic sectors to establish new regulations and improve the existing ones are needed. In this sense, it should be a reality in the near future that there are more initiatives to certify the characteristics of industrial products which are standardized worldwide and that there are key standardization bodies to legislate and certify products with logos that describe the properties of the family of bioplastics.

## 8. Conclusions

The field of plastics has been changing over the years, with biobased molecules originally made to be synthetic, complex, and nonbiodegradable. The stability conferred by the polymeric nature of traditional plastics hinders their degradation in the environment, resulting in the accumulation of plastics in the ecosystem instead of their chemical degradation, macroscopic fragmentation, and biological assimilation. Research in the plastic field is overwhelmingly focused on tackling its environmental and economic problems. Trends in the use of oxodegradable additives and their potential degradation by microbial and physical means are an insight into these topics. Nevertheless, the expectation to use different raw materials derived from natural sources and create a circular economy to make biopolymers that substitute for fossil resources to produce plastics is on the rise. Although the physicochemical properties of the materials could be exceptional in relation to biodegradability, challenges derived from their production, implementation, and recyclability have been identified. The relationship between the activities of (1) governments with public policy, (2) academics with research on new materials and their properties, (3) the industry with the implementation and products, (4) international organizations and nonprofit organizations that supply regulations and certifications, and (5) the social culture regarding the use of plastics and recyclability will bring better initiatives for the future of plastics.

## Figures and Tables

**Figure 1 polymers-14-01203-f001:**
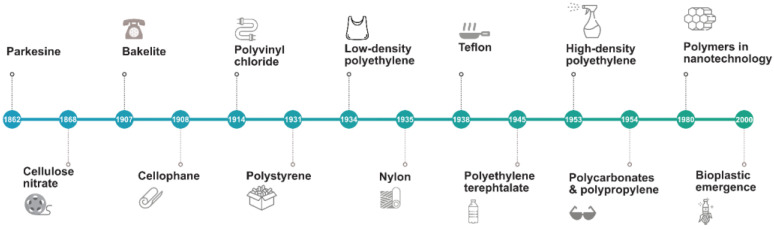
Timeline of plastics from 19th to 21st century. Designed online at Freepik.com [Last accessed 23 February 2022].

**Figure 2 polymers-14-01203-f002:**
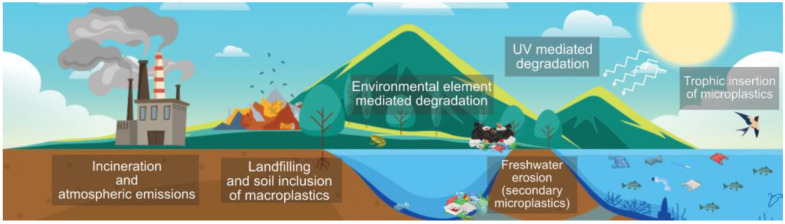
Insertion pathways of plastic-related pollutants into the environment. Designed online at flaticon.com [Last accessed: 23 February 2022].

**Figure 3 polymers-14-01203-f003:**
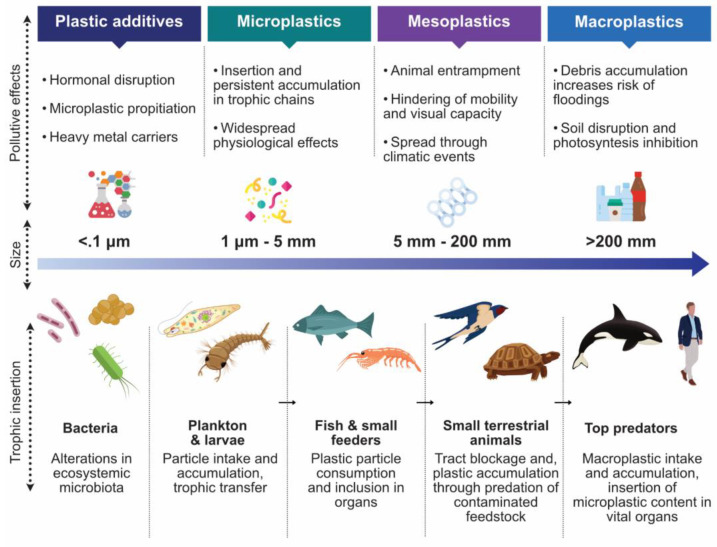
Size, features, and related effects of plastic-derived pollutants in the trophic chain. Designed online at flaticon.com [Last accessed: 23 February 2022].

**Figure 4 polymers-14-01203-f004:**
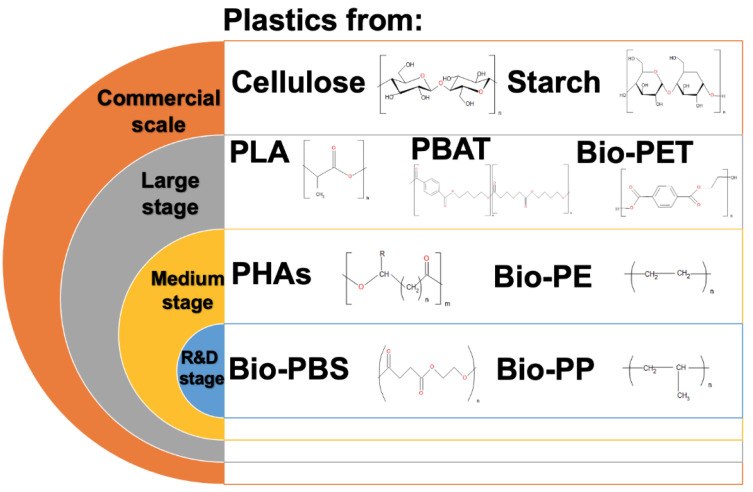
Stages of bioplastic production. PLA—polylactic acid; PBAT—poly(1,4-butylene adipate-co-1,4-butylene terephthalate); Bio-PET—bio-poly(ethylene terephthalate); PHAs—polyhydroxyalkanoates); Bio-PE—bio-polyethylene; Bio-PBS—bio-polybutylene succinate; Bio-PP—bio-polypropylene.

**Figure 5 polymers-14-01203-f005:**
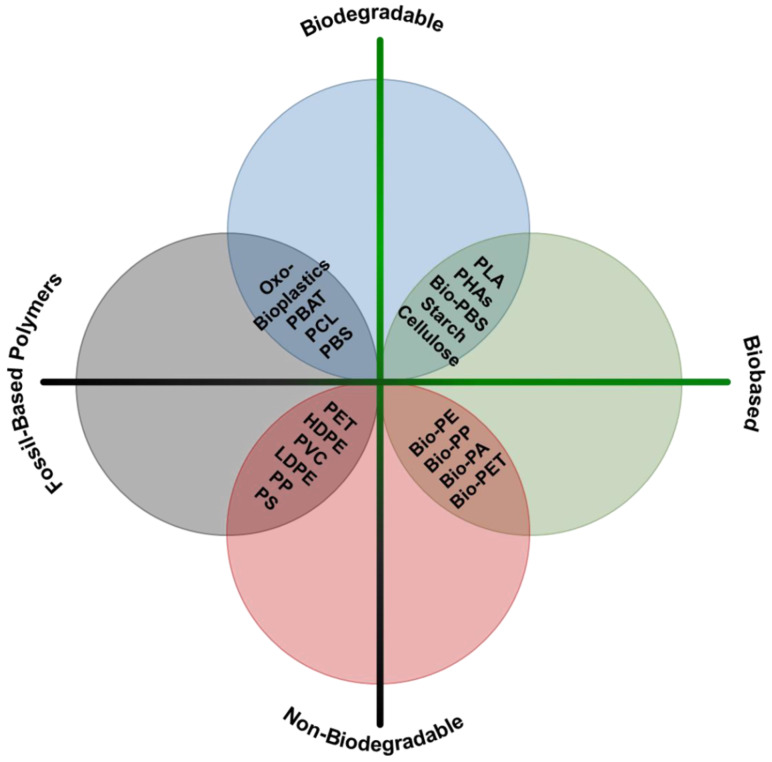
Scheme of bioplastic classification. PLA—polylactic acid; PHAs—polyhydroxyalkanoates; Bio-PBS—bio-polybutylene succinate; Bio-PE—bio-polyethylene; Bio-PP—bio-polypropylene; Bio-PA—bio-polyamide; Bio-PET—bio-poly(ethylene terephthalate); PET—poly(ethylene terephthalate); HDPE—high-density polyethylene; PVC—polyvinyl chloride; LDPE—low-density polyethylene; PP—polypropylene; PS—polystyrene; PBAT—poly(1,4-butylene adipate-co-1,4-butylene terephthalate); PCL—polycaprolactone; PBS—polybutylene succinate.

**Figure 6 polymers-14-01203-f006:**
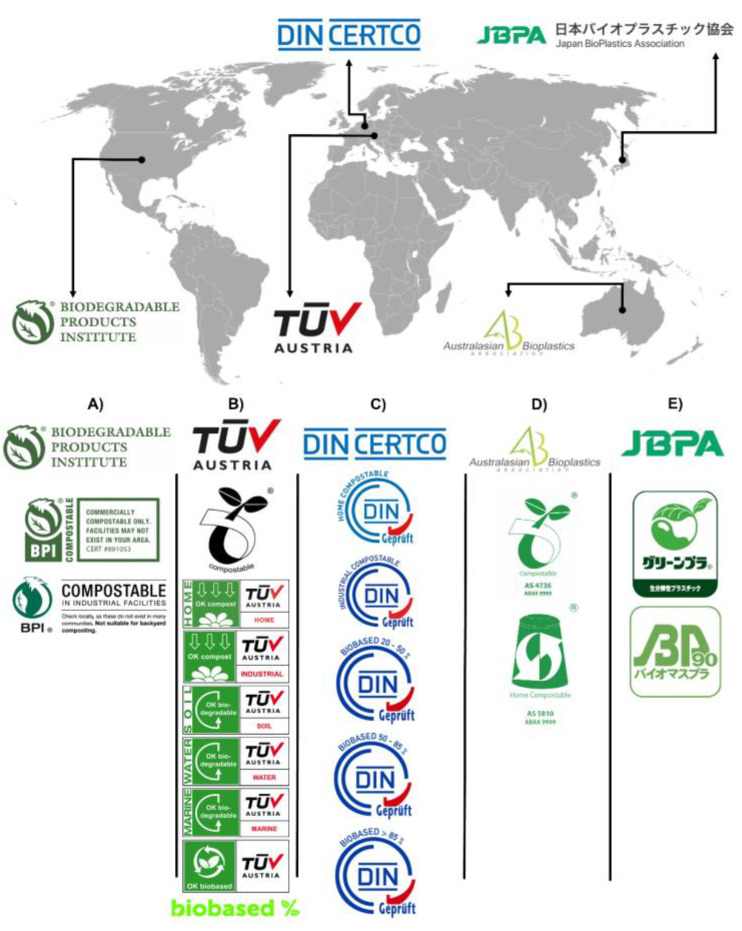
Bioplastic certification world map. (**A**) Biodegradable Product Institute; (**B**) Tüv Austria; (**C**) DIN Certco; (**D**) Australasian Bioplastics Association; (**E**) Japan BioPlastics Association.

**Table 1 polymers-14-01203-t001:** Biopolymer production from plants.

Biopolymer	Source	Reinforcement	Plasticizer	Reference
Starch	Corn and cassava	*Cola cordifolia*	Glycerol	[127]
	Rice and corn	Ethanol, rice, and olive oil	Sorbitol	[128]
	Tapioca	Sugarcane bagasse fiber	Glycerol	[129]
	Banana peel		Glycerol	[130]
	Corn, potato, and cassava	Recycled newspaper pulp fiber	Glycerol	[131]
	Cassava	Microcrystalline cellulose	Sorbitol	[132]
	Tapioca		Acetyl Tributyl Citrate	[133]
	Corn	Microalgae *Nannochloropsis*	Glycerol	[134]
	Microalgae *Spirulina* sp.	Poly(vinyl alcohol)		[135]
	Microalgae *Chlamydomonas reinhardtii* 11-32A		Glycerol	[136]
Cellulose acetate	Cotton linters		Polyethylene glycol 600	[137]
	Flax fibers		Polyethylene glycol 600	[137]
	*Parthenium hysterophorus* weed		Polyethylene glycol 600	[138]

**Table 3 polymers-14-01203-t003:** Commercial applications of biobased polymers and their properties.

Biopolymer	Applications	Properties	Cost USD/kg	Reference
Starch	Translucent film, net packaging, bags, containers, egg boxes, sandwich bags, capsules, carrier bags, drinking straws, drug-release films	Sealable, durable, fine finishing, barrier for water	0.5–2.0	[118]
Cellulose	Packaging films, films, transparent films, barrier films, cups for cold drinks, plates and dishes, cups for hot drinks, labels	Sealable, barrier for water, transparent, approved for direct food contact	1.8–4.0	[118]
PLA ^1^	Bottles, cups, transparent films, containers, dishes, fruit nets, top-covering films, trays, tea bags, ice cream cups, carrier bags	Approved for direct contact, transparent, sealable, durable, barrier for water and oxygen	4.0–6.0	[118]
PHA ^2^	Disposable cups, plates, and cutlery; Tetra Pak covers; tubes to produce vegetable seedlings; agrochemical packaging; textile fibers; electronic equipment components	Physical properties like conventional plastics; insoluble in water, nontoxic, and biocompatible; present piezoelectric properties; some PHA films exhibit gas-barrier properties	2.4–5.5	[157]
Bio-PE ^3^	Food packaging, cosmetics, personal care, automotive and toy applications	Equal in its chemical, physical, and mechanical properties to fossil-based PE	2.3	[115]
PBS ^4^	Biopackaging, tissue-engineering, and medical materials; agriculture mulch film; plant pots; hygiene products	High processability, good mechanical properties, thermal properties	4.0–10.0	[120,178]
PLC ^5^	Drug delivery systems and tissue-engineering scaffolds	High toughness and flexibility, biocompatibility, and slow degradation in in vivo conditions	4.5–10.0	[120]
PBAT ^6^	Compostable organic waste bags, agricultural mulch films, packaging (wrapping) films, disposable tableware	Excellent toughness, improved wear and fracture resistance, good chemical resistance to water and oils, high strain at break	3.8–5.8	[120]

^1^ PLA—polylactic acid; ^2^ PHA—polyhydroxyalkanoates ^3^ Bio-PE—biobased polyethylene; ^4^ PBS—polybutylene succinate; ^5^ PCL—polycaprolactone; ^6^ PBAT—polybutylene adipate terephthalate.

**Table 4 polymers-14-01203-t004:** Commercial bioplastics and business cases.

Company	Bioplastic	Applications	Properties	Country
Plantic^®® 1^	Starch	Food and goods packing, agricultural plastics	Biodegradable and compostable	Australia
Mater-Bi^®®^- Novamont ^2^	Starch	Bags, toys, food, and cosmetic containers	Biodegradable and compostable	Italy
BIOPAR^®® 3^	Starch	Bags and flexible packaging	Biodegradable	Portugal
Biofase^®® 4^	Starch-based	Cutlery	Biodegradable	Mexico
Solany^®® 5^	Starch-derived	Flowerpots, tomato clips, cultivation tubes, promotional items, toys, CD and DVD trays, protection covers for packaging, cup holders, plant stakes, golf tees	Biodegradable and compostable	Canada
Bionolle Starcla^TM^ -Showa Denko ^6^	Starch- and PLA-based	Bioplastics	Biodegradable and compostable	Japan
BIOFRONT-Teijin ^7^	Stereocomplex PLA ^13^	Automotive, films and packaging, molded parts for civil engineering and construction, parts for electronic devices	Biodegradable	Japan
Ingeo^TM^-Nature Works ^8^	PLA ^13^	Bottles, gift cards, durable goods, films, layers of paper, cups and containers for food, fabrics, clothing, disposables, and base material for many compounds	Biodegradable and compostable	USA
WeforYou ^9^	PLA ^13^	Reusable bags	Biodegradable and compostable	Austria
Total-Corbion ^10^	PLA ^13^	Biopolymer	Biodegradable and compostable	Netherlands/ Thailand
Danimer Scientific ^11^	PHA ^14^	Straws, cups, lids, bottles, produce bags, shopping bags, cutlery, diaper linings, plates, wipes, toys, trash bags, seals, labels, glues, and much more	Biodegradable and compostable	USA ^15^
Qmilk ^12^	Milk protein	Textile fibers	Compostable	Germany

^1^ Website: https://plantic.com.au/ (accessed on 3 February 2022); ^2^ Website: https://materbi.com/ (accessed on 3 February 2022); ^3^ Website: https://unitedbiopolymers.com/ (accessed on 3 February 2022); ^4^ Website: https://biofase.com.mx/ (accessed on 3 February 2022); ^5^ Website: https://solanylbiopolymers.com/ (accessed on 3 February 2022); ^6^ Website: https://www.sdk.co.jp/ (accessed on 3 February 2022); ^7^ Website: https://www.teijin.co.jp/ (accessed on 03 February 2022); ^8^ Website: https://www.natureworksllc.com/ (accessed on, 3 February 2022); ^9^ Website: https://weforyou.pro/ (accessed on 3 February 2022); ^10^ Website: https://www.total-corbion.com/ (accessed on, 3 February 2022); ^11^ Website: https://danimerscientific.com/ (accessed on 3 February 2022); ^12^ Website: https://www.qmilkfiber.eu/ (accessed on 3 February 2022). ^13^ PLA—Polylactic acid; ^14^ PHA—Polyhydroxyalkanoates; ^15^ USA—United States of America.

**Table 5 polymers-14-01203-t005:** Chemical and biological strategies for degradation.

Plastic	Biodegradation Conditions (Chemical/Biological)	Biodegradation	Reference
Cassava-based bioplastic	Burial-soil pH measurement, 14 days (using microorganisms)		[127]
Starch (TPS)–PLA ^1^	*Ulomoides dermestoides*, 5 days	TPS biodigestion—biodegradation (80%) and PLA biodisintegration (50%)	[186]
PHA ^2^	Alluvial-type soil, 35% soil moisture, 60 days	35%	[187]
HDPE ^3^	Incubation with microbial consortium, 357 days	15%	[188]
LDPE ^4^		4.96%	
PP ^5^		6.7%	
PS ^6^		5.29%	
	Incubation under standard test aerobic and anaerobic conditions	Aerobic conditions, 117 days	[189]
PHB ^7^		PHB ^7^ 83%	
		PBHV ^8^ 87.4%	
PHBV ^8^		PCL ^10^ 77.6%	
PBS ^9^		Anaerobic, 77 days	
PCL ^10^		PHB ^7^ 83.9%	
PLA ^1^		PBHV ^8^ 81.2%	
PET ^11^	PET^7^ hydrolase enzyme, 10 h	90%	[190]
PET ^11^	Recombinant bacterial polyester hydro- lase TfCut2, expressed in *Bacillus subtilis*, 70 °C, 96 h	50%	[191]
LDPE ^2^ and HDPE ^3^	Thermal degradation (pyrolysis), 30 to 550 °C at 5 °C min^−1^	1-oleofins and n-paraffins if C2–C6 were the major products	[192]
LDPE ^2^ and PS ^6^	Pyrolysis, 300–500 °C, nitrogen pressure of 0.3 MPa	LDPE ^2^ was degraded to oil at 425 °C PS ^6^ was degraded at around 350 °C	[193]
PS ^6^	Pyrolysis, room temperature 800 °C under inert atmosphere	70%	[194]
Polyethylene (HDPE) ^3^ pellets	Thermal pyrolysis, 350 °C	81%; the oil consisted mainly of paraffinic hydrocarbons, most of which contained between 6 and 16 carbon atoms	[195]
Cellulose	Enzymatic degradation (endoglucanases, β-glucosidases, endoxylanases, β-xylosidases, mannosidases), 7 days	0.5% (*w/v*)	[196]
PCL ^10^	Enzymatic degradation (external PCL ^5^ depolymerase), 10 days	>80%	[197]

^1^ PLA—polylactic acid, ^2^ PHA—polyhydroxyalkanoate, ^3^ HDPE—high-density polyethylene; ^4^ LDPE—low-density polyethylene; ^5^ PP—polypropylene; ^6^ PS—polystyrene; ^7^ PHB—poly(3-hydroxybutyrate-co-3-hydroxyvalerate); ^8^ PHBV—poly(butylene succinate); ^9^ PBS—poly(butylene adipate-co-terephthalate); ^10^ PCL—poly(ε-caprolactone); ^11^ PET—polyethylene terephthalate.

**Table 6 polymers-14-01203-t006:** Standards related to bioplastics from selected countries.

Country	Nomenclature of the Standard	Title of Standard
Mexico	NMX-E-273-NYCE-2019	Plastic Industry—Compostable plastics—Specifications and essay methods
	NMX-E-267-CNCP-2016	Plastic industry—Biobased plastics—Essay methods
USA	ASTM D5071-06(2013)	Standard practice for exposure of photodegradable plastics in a xenon arc apparatus
	ASTM D5208-14	Standard practice for fluorescent ultraviolet (UV) exposure of photodegradable plastics
	ASTM D5272-08(2013)	Standard practice for outdoor exposure testing of photodegradable plastics
	ASTM D5338-15	Standard test method for determining aerobic biodegradation of plastic materials under controlled composting conditions, incorporating thermophilic temperatures
	ASTM D5511-18	Standard test method for determining anaerobic biodegradation of plastic materials under high-solids anaerobic-digestion conditions
	ASTM D5526-18	Standard test method for determining anaerobic biodegradation of plastic materials under accelerated landfill conditions
	ASTM D5988-18	Standard test method for determining aerobic biodegradation of plastic materials in soil
	ASTM D6400-19	Standard specification for labeling of plastics designed to be aerobically composted in municipal or industrial facilities
	ASTM D6691-17	Standard test method for determining aerobic biodegradation of plastic materials in the marine environment by a defined microbial consortium or natural sea water inoculum
	ASTM D6866-21	Standard test methods for determining the biobased content of solid, liquid, and gaseous samples using radiocarbon analysis
	ASTM D6868-21	Standard specification for labeling of end items that incorporate plastics and polymers as coatings or additives with paper and other substrates designed to be aerobically composted in municipal or industrial facilities
	ASTM D6954-18	Standard guide for exposing and testing plastics that degrade in the environment by a combination of oxidation and biodegradation
	ASTM D7444-18a	Standard practice for heat and humidity aging of oxidatively degradable plastics
	ASTM D7475-20	Standard test method for determining the aerobic degradation and anaerobic biodegradation of plastic materials under accelerated bioreactor landfill conditions
	ASTM D7991-15	Standard test method for determining aerobic biodegradation of plastics buried in sandy marine sediment under controlled laboratory conditions
UK	BS 8472:2011	Methods for the assessment of the oxo-biodegradation of plastics and of the phyto-toxicity of the residues in controlled laboratory conditions
	BS ISO 16620-1:2015	Plastics. Biobased content General principles
	BS ISO 16620-2:2019	Plastics. Biobased content Determination of biobased carbon content
	PD CEN/TR 16721:2014	Biobased products. Overview of methods to determine the biobased content (British standard)
	BS ISO 16620-3:2015	Plastics. Biobased content Determination of biobased synthetic polymer content
	BS ISO 22526-3:2020	Plastics. Carbon and environmental footprint of biobased plastics Process carbon footprint, requirements, and guidelines for quantification
	BS ISO 23517:2021	Plastics. Soil biodegradable materials for mulch films for use in agriculture and horticulture. Requirements and test methods regarding biodegradation, ecotoxicity and control of constituents
	BS ISO 5412	Biodegradable plastic shopping bags for industrial composting
EU	CSN EN ISO 10210	Plastics—Methods for the preparation of samples for biodegradation testing of plastic materials
	DIN EN 13432	Requirements for packaging recoverable through composting and biodegradation—Test scheme and evaluation criteria for the final acceptance of packaging
	CSN EN ISO 14851	Determination of the ultimate aerobic biodegradability of plastic materials in an aqueous medium—Method by measuring the oxygen demand in a closed respirometer
	CSN EN ISO 14852	Determination of the ultimate aerobic biodegradability of plastic materials in an aqueous medium—Method by analysis of evolved CO_2_
	CSN EN ISO 14853	Plastics—Determination of the ultimate anaerobic biodegradation of plastic materials in an aqueous system—Method by measurement of biogas production
	CSN EN ISO 14855-1	Determination of the ultimate aerobic biodegradability of plastic materials under controlled composting conditions—Method by analysis of evolved CO_2_—Part 1: General method
	CSN EN ISO 14855-2	Determination of the ultimate aerobic biodegradability of plastic materials under controlled composting conditions—Method by analysis of evolved CO_2_—Part 2: Gravimetric measurement of CO_2_ evolved in a laboratory-scale test
	CSN EN 14995	Plastics—Evaluation of compostability—Test scheme and specifications
	CSN EN ISO 15985	Plastics—Determination of the ultimate anaerobic biodegradation under high-solids anaerobic-digestion conditions—Method by analysis of released biogas
	CSN EN 16640	Biobased products—Biobased carbon content—Determination of the biobased carbon content using the radiocarbon method
	CSN EN 16760	Biobased products—Life Cycle Assessment
	EN 16785-1	Biobased products—Biobased content—Part 1: Determination of the biobased content using the radiocarbon analysis and elemental analysis
	CSN EN 16785-2	Biobased products—Biobased content—Part 2: Determination of the biobased content using the material balance method
	CSN EN ISO 16929	Plastics—Determination of the degree of disintegration of plastic materials under defined composting conditions in a pilot-scale test
	CSN EN ISO 17556	Plastics—Determination of the ultimate aerobic biodegradability of plastic materials in soil by measuring the oxygen demand in a respirometer or the amount of CO_2_ evolved
	CSN EN 17417	Determination of the ultimate biodegradation of plastics materials in an aqueous system under anoxic (denitrifying) conditions—Method by measurement of pressure increase
	CSN EN ISO 18830	Plastics—Determination of aerobic biodegradation of nonfloating plastic materials in a seawater/sandy sediment interface—Method by measuring the oxygen demand in closed respirometer
	CSN EN ISO 19679	Plastics—Determination of aerobic biodegradation of nonfloating plastic materials in a seawater/sediment interface—Method by analysis of evolved CO_2_
International	ISO 14851	Determination of the ultimate aerobic biodegradability of plastic materials in an aqueous medium—Method by measuring the oxygen demand in a closed respirometer
	ISO 14852	Determination of the ultimate aerobic biodegradability of plastic materials in an aqueous medium—Method by analysis of evolved CO_2_
	ISO 14853	Plastics—Determination of the ultimate anaerobic biodegradation of plastic materials in an aqueous system—Method by measurement of biogas production

## Data Availability

Not applicable.

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
