# Peer review of "Towards a Circular Economy of Plastics: An Evaluation of the Systematic Transition to a New Generation of Bioplastics"

_polymers, 2022, doi:10.3390/polym14061203_

Round 1

Reviewer 1 Report

The present manuscript describes an in-depth analysis of the data published regarding especially of bioplastics, but the history and environmental impact of plastics together with the recent bioplastic trends, production, and legislations are also included. I suggest major revisions as there are many issues (of correctness, quality of presentation, and English language), as described in detail in the following.

Required corrections:

  • The title is a little bit misleading.

The article also tackles the environmental effects of plastics; this should somehow also be integrated into the title.

  • line 76 – 78 – Some additional references are needed (i.e. https://doi.org/10.3390/polym13203574, https://doi.org/10.1007/s11356-021-17792-w)
  • line 129 – please revise this section “Nylon was first formulated in 1933 as a sticky material with slight structural integrity.” which comes after the statement “After World War II, new materials emerged.” – actually I think that nylon was formulated before the WWII, and “Polyamide 66 and Teflon were discovered in 1941.” – which was right in the period of WWII.
  • line 137 – please correct in temperatures to at temperatures
  • line 251 - Please cite this recent relevant reference: https://doi.org/10.3390/coatings12010102
  • An important reference: https://doi.org/10.1016/j.tibtech.2019.04.011
  • e. line 450 and line 453 - I think that the authors should use one format for carbon dioxide (CO2)
  • line 465-467 - A great overview regarding the family of bioplastics can be found in:
  • . https://doi.org/10.3390/su13147848
  • table 2 - Maybe some terms could be abbreviated Batch bioreactor BB, Shake flask SF, and so on for a better overview of the table
  • line 543 – “as PET and PS It is biocompatible,” please correct
  • line 604 and table 3 - in the table the authors use USD, they should be consistent and use it the same way through the article
  • line 629 – please correct “utensils”
  • table 4 - please define every abbreviation in the table footer
  • line 646 – 651 - please insert the accession date for each site
  • line 666 – please rephrase and revise in the whole document “biodegradable and compostable biopolymers”
  • line 685 – please correct “prospective” to plural
  • line 699 – 700 - please rephrase it is hard to understand (the same at line 744-745, 765-769), also revise this section carefully

Overall, the manuscript is good, but from my perspective, it should be a little bit reorganized and some sections should be rephrased as they are hard to follow. The English should be revised, please carefully proofread, spell check to eliminate grammatical errors for clarity and correctness.

Author Response

We thank you for the detailed review and useful comments. We have done extensive language editing throughout the manuscript. We have followed your suggestion and accordingly revised the manuscript. The changes made in the revised manuscript are highlighted in GREEN color and shown in this document.  Please note our responses to your comments.

We thank you for the detailed review and useful comments. We have done extensive language editing throughout the manuscript. We have followed your suggestion and accordingly revised the manuscript. The changes made in the revised manuscript are highlighted in GREEN color and shown in this document.  Please note below our responses to your comments.

The present manuscript describes an in-depth analysis of the data published regarding especially of bioplastics, but the history and environmental impact of plastics together with the recent bioplastic trends, production, and legislations are also included. I suggest major revisions as there are many issues (of correctness, quality of presentation, and English language), as described in detail in the following.

Author´s response: Thank you for your valuable comment. We have done extensive language editing throughout the manuscript. We have followed your suggestions accordingly.

Required corrections:

The title is a little bit misleading.

The article also tackles the environmental effects of plastics; this should somehow also be integrated into the title.

Author´s response: Thank you for your valuable comment. The title was improved to tackle the circular economy of plastics.

Towards a circular economy of plastics: An evaluation of the systematic transition to a new generation of bioplastics

line 76 – 78 – Some additional references are needed (i.e. https://doi.org/10.3390/polym13203574, https://doi.org/10.1007/s11356-021-17792-w)

Author´s response: Thank you for your valuable comment. The references were added in the introduction and  section 6.

line 129 – please revise this section “Nylon was first formulated in 1933 as a sticky material with slight structural integrity.” which comes after the statement “After World War II, new materials emerged.” – actually I think that nylon was formulated before the WWII, and “Polyamide 66 and Teflon were discovered in 1941.” – which was right in the period of WWII.

Author´s response: Thank you for your valuable comment. We have addressed this issue. Please see line 125-127.

line 137 – please correct in temperatures to at temperatures

Author´s response: Thank you very much for your suggestion. We have addressed this issue

line 251 - Please cite this recent relevant reference: https://doi.org/10.3390/coatings12010102

Author´s response: Thank you for your valuable comment. The references were added in section 5.2.

An important reference: https://doi.org/10.1016/j.tibtech.2019.04.011

  1. line 450 and line 453 - I think that the authors should use one format for carbon dioxide (CO2)

Author´s response: Thank you for your comment. The format of carbon dioxide was homogenized to CO2 in the whole document. The reference was added in a new section 5.2

line 465-467 - A great overview regarding the family of bioplastics can be found in:

. https://doi.org/10.3390/su13147848

Author´s response: Thank you for your suggestion. Figure 5 was improved according the reference 119.

table 2 - Maybe some terms could be abbreviated Batch bioreactor BB, Shake flask SF, and so on for a better overview of the table

Author´s response: Thank you very much for your suggestion. The abbreviations were added for a better overview. Please Table 2.

line 543 – “as PET and PS It is biocompatible,” please correct

Author´s response: Thank you very much for your suggestion. We have addressed this issue. Please see line 546.

line 604 and table 3 - in the table the authors use USD, they should be consistent and use it the same way through the article

Author´s response: Thank you for valuable comment. We have unified as US in the whole document.

line 629 – please correct “utensils”

Author´s response: Thank you very much for your suggestion. We have addressed this issue. Please see line 636

table 4 - please define every abbreviation in the table footer

Author´s response: Thank you for your suggestion. The abbreviations were added. Please see Table 4

line 646 – 651 - please insert the accession date for each site

Author´s response: Thank you for your suggestion. The date of accession was added. Please see Table 4

line 666 – please rephrase and revise in the whole document “biodegradable and compostable biopolymers”

Author´s response: Thank you very much for your suggestion. We have addressed this issue. Please see line 679.

line 685 – please correct “prospective” to plural

Author´s response: Thank you very much for your suggestion. We have addressed this issue. Please see line 714

line 699 – 700 - please rephrase it is hard to understand (the same at line 744-745, 765-769), also revise this section carefully

Author´s response: Thank you for noticing it. We have reconstructed the paragraphs according to your suggestion. Please see line, 725-726, 743-746, and 773-774.

Overall, the manuscript is good, but from my perspective, it should be a little bit reorganized and some sections should be rephrased as they are hard to follow. The English should be revised, please carefully proofread, spell check to eliminate grammatical errors for clarity and correctness.

Author´s response: Thank you for your valuable comment. We have done extensive language editing throughout the manuscript

Reviewer 2 Report

The presented review is devoted to an important and challenging problem of plastic management. Currently, plastics waste presents a huge environmental problem contaminating land and water all over the Globe. The Authors comprehensively covered petro-based and bio-based plastics, sources for their production and economics of it, as well as feasibility of their degradation. The only “blind spot” that has not been covered, at least in some details, is biodegradation. While a table is devoted to microorganisms involved in plastics bio-production, not much was said about microorganisms involved in biodegradation and research devoted to accelerating this process. Otherwise, the review provides lots of valuable information and a very useful read.

I have some comments on style/grammar mostly.

Line 97. Instead of “alarming indicators”, it’s better to use “despite some concerns exist”…

Line 101. “ During the last centuries” , remove “last” as you are going to antique times.

Line 103. Instead “uses”, use “applications”

Lines 111-112. Instead of “exhibited”, use “demonstrated. Then use “prepared by dissolution ..”

Line 114. “American” does not indicate a country. It’s better to say “development in the US”

Line 117. It’s better to write “in demand”. Next sentence should be re-phrased as “ … a first proteineous member of natural polymers family”

Line 133. Insert an article “the” in front of US

Line 135 “higher melting” miss a word “temperature”

Line 138. Use “invented or presented “ instead of “prepared”

Line 140. Remove “of” in front of 1990

Line 148. PP -first time mentioned- should use full name

Line 229. Use “bacterial consortia”

Line 332. Is it “bisphenol A” or biphenol A. Check, please.

Line 333. “lessons” is a wrong word here. Check, please.

Line 397 “biotical” is a wrong term. Use “biological”, please.

Line 398. Remove semi-colon. Finish the sentence and start a new one.

Line 404-405. “Known to contain no more than 49 atoms of carbon, a potential substrate for microorganism transformation [105]. “ This is a wrong statement. 49 carbon atoms do not guarantee that a compound is biodegradable or bio-transformable.

Line 414. “the feasibility of oxo-plastics” is a bad phrasing. Should be changed.

Figure 3. It could be easier for a reader if in a legend full name for plastics abbreviations were provided.

Line 492. “it has been studied to mix” is a bad wording. It’s better to use “attempts have been made”..

Line 494. Use “Packaging” instead of “packing”.

Line 510. The same as above

Line 514. It’s better to say “more pure” than “less impure”

Line 521. Introducing Table 2, say something that data summarized not only bacterial cellulose but about other polymers too.

Line 556 “Cobs” plural

Line 556. “Even when PLA has been  extensively accepted”, it is better to say Despite that PLA has been…”

Line 562-563. “they are accumulated by multiple bacteria” is redundant. Should be removed.

Line 597-598. “Crab shells and tree fiber were used to  contain liquids and foods, and their composition was chitin and cellulose [167]. “ Bad phrase. Re-phrase, please.

Line 606. Change “for” to “to”.

Line 634. Add a phrase to better introduce Table 3.

Table 4. Change column title from “uses” to “applications”

Line 670. Remove “other”

Line 698. Use “storage” instead of “store”

Line 699-700. Use “PLA production optimization requires a selection of microorganism to increase yield”

Line 712. “Hydrolysis” is just one of mechanism of biodegradation. In general case, like here we do not know what mechanisms biodegradation will proceed. So, it is better to use a general term “biodegradation”

Line 716. “..plastic by organic materials such as resins, additives or thin films to promote their biodegradability is a common limiting factor for their recyclability” This statement is confusing. If a thing is biodegradable, how can it limit their recyclability? Requires clarification

Line 731. Change “these” to “they”

Line 740-741. The phrase starting with “Although the current report deal with….” is very awkward and has to be re-phrased in a way that a reader could understand that burning is not a very good option as bio-produced and petro-produced polymers will contribute the same amount of CO2 to the atmosphere.

Line 743. Change “troubles” to “problems”

Line 744. Remove “into” in front of “have gone”. Change “de” to “a”. Remove dot after “biodegradability”and put a commar.

Line 765. Put “such” in front of “as”.

Line 766. Remove “of great interest”

Line 775. I suggest removing “social problem”, because then you need explain it .

Author Response

We thank you for the detailed review and useful comments. We have done extensive language editing throughout the manuscript. We have followed your suggestion and accordingly revised the manuscript. The changes made in the revised manuscript are highlighted in GREEN color and shown in this document.  Please note below our responses to your comments.

Reviewer 2

The presented review is devoted to an important and challenging problem of plastic management. Currently, plastics waste presents a huge environmental problem contaminating land and water all over the Globe. The Authors comprehensively covered petro-based and bio-based plastics, sources for their production and economics of it, as well as feasibility of their degradation. The only “blind spot” that has not been covered, at least in some details, is biodegradation. While a table is devoted to microorganisms involved in plastics bio-production, not much was said about microorganisms involved in biodegradation and research devoted to accelerating this process. Otherwise, the review provides lots of valuable information and a very useful read.

Author´s response: Thank you for your valuable comment. Table 5 was added to address topics on chemical and biological strategies for degradation of plastics.

I have some comments on style/grammar mostly.

Line 97. Instead of “alarming indicators”, it’s better to use “despite some concerns exist”…

Author´s response: Thank you for your valuable comment. The change was done. Please see line 92

Line 101. “ During the last centuries” , remove “last” as you are going to antique times.

Author´s response: Thank you for your valuable comment, the line was reconstructed. Please see line 95

Line 103. Instead “uses”, use “applications”

Author´s response: Thank you for your valuable comment, the word was changed.

Lines 111-112. Instead of “exhibited”, use “demonstrated. Then use “prepared by dissolution ..”

Author´s response: Thank you for your suggestion. The word was modified. Please see line 105.

Line 114. “American” does not indicate a country. It’s better to say “development in the US”

Author´s response: Thank you for your suggestion. The change was done. Please see line 108.

Line 117. It’s better to write “in demand”. Next sentence should be re-phrased as “ … a first proteineous member of natural polymers family”

Author´s response: Thank you for your suggestion. We have reconstructed the paragraph. Please see line 109.

Line 133. Insert an article “the” in front of US

Author´s response: Thank you for your suggestion. The change was done. Please see line 108.

Line 135 “higher melting” miss a word “temperature”

Author´s response: Thank you for your suggestion. The change was done. Please see line 132.

Line 138. Use “invented or presented “ instead of “prepared”

Author´s response: Thank you for your suggestion. The change was done

Line 140. Remove “of” in front of 1990

Author´s response: Thank you for your valuable comment, the word was removed. Please see line 191.

Line 148. PP -first time mentioned- should use full name

Author´s response: Thank you for your suggestion. We added the full name. Please see line 146.

Line 229. Use “bacterial consortia”

Author´s response: Thank you for your suggestion. The change was done. Please see line 232.

Line 332. Is it “bisphenol A” or biphenol A. Check, please.

Author´s response: Thank you for your valuable comment. The correct expression is bisphenol A, it was consulted in pubchem.

https://pubchem.ncbi.nlm.nih.gov/compound/bisphenol-A

Line 333. “lessons” is a wrong word here. Check, please.

Author´s response: Thank you for your valuable comment. The word “lessons” was changed to lesions. Please see line 338.

Line 397 “biotical” is a wrong term. Use “biological”, please.

Author´s response: Thank you for your suggestion. The change was done. Please see line 399

Line 398. Remove semi-colon. Finish the sentence and start a new one.

Author´s response: Thank you for your suggestion. The comment was addressed.

Line 404-405. “Known to contain no more than 49 atoms of carbon, a potential substrate for microorganism transformation [105]. “ This is a wrong statement. 49 carbon atoms do not guarantee that a compound is biodegradable or bio-transformable.

Author´s response: Thank you for your constructive comments. The reference was replaced to extend the discussion on the use of microorganisms on biodegradation process of polyethylene oligomers. Please see line 404.

Line 414. “the feasibility of oxo-plastics” is a bad phrasing. Should be changed.

Author´s response: Thank you for your constructive comments. We improved the redaction of the paragraph. Please see line 416-417.

Figure 3. It could be easier for a reader if in a legend full name for plastics abbreviations were provided.

Author´s response: Thank you for your suggestion. The legend full names were added for more details and easier reader. Please see Figure 4.

Line 492. “it has been studied to mix” is a bad wording. It’s better to use “attempts have been made”..

Author´s response: Thank you for your suggestion. The change was done. Please see line 499.

Line 494. Use “Packaging” instead of “packing”.

Author´s response: Thank you for your suggestion. The change was done. Please see line 501.

Line 510. The same as above

Author´s response: Thank you for your suggestion. The change was done. Please see line 516.

Line 514. It’s better to say “more pure” than “less impure”

Author´s response: Thank you very much for your suggestion. We have addressed this issue. Please see line 521.

Line 521. Introducing Table 2, say something that data summarized not only bacterial cellulose but about other polymers too.

Author´s response: Thank you for your valuable comment. The table was moved to below to introduce before all the polymers.

Line 556 “Cobs” plural

Author´s response: Thank you very much for your suggestion. We have addressed this issue. Please see line 559.

Line 556. “Even when PLA has been  extensively accepted”, it is better to say Despite that PLA has been…”

Author´s response: Thank you very much for your suggestion. The comment was attended. Please see line 559.

Line 562-563. “they are accumulated by multiple bacteria” is redundant. Should be removed.

Author´s response: Thank you for your suggestion. The change was done

Line 597-598. “Crab shells and tree fiber were used to contain liquids and foods, and their composition was chitin and cellulose [167]. “ Bad phrase. Re-phrase, please.

Author´s response: Thank you for your suggestion. We have addressed this issue and reconstructed the phrase. Please see line 603.

Line 606. Change “for” to “to”.

Author´s response: Thank you for your suggestion. The change was done

Line 634. Add a phrase to better introduce Table 3.

Author´s response: Thank you for your suggestion. The phrase was added to introduce Table 3.

Table 4. Change column title from “uses” to “applications”

Author´s response: Thank you for your suggestion. The change was done

Line 670. Remove “other”

Author´s response: Thank you for your suggestion. The word was removed.

Line 698. Use “storage” instead of “store”

Author´s response: Thank you for your suggestion. We have addressed this issue. Please see line 728.

Line 699-700. Use “PLA production optimization requires a selection of microorganism to increase yield”

Author´s response: Thank you for your constructive comments. The paragraph was improved. Please see line 728.

Line 712. “Hydrolysis” is just one of mechanism of biodegradation. In general case, like here we do not know what mechanisms biodegradation will proceed. So, it is better to use a general term “biodegradation”

Author´s response: Thank you for your valuable comment. The change was done

Line 716. “..plastic by organic materials such as resins, additives or thin films to promote their biodegradability is a common limiting factor for their recyclability” This statement is confusing. If a thing is biodegradable, how can it limit their recyclability? Requires clarification

Author´s response: Thank you for your valuable comment. The paragraph was reconstructed. Please see line 748.

Line 731. Change “these” to “they”

Author´s response: Thank you for your suggestion. The change was done

Line 740-741. The phrase starting with “Although the current report deal with….” is very awkward and has to be re-phrased in a way that a reader could understand that burning is not a very good option as bio-produced and petro-produced polymers will contribute the same amount of CO2 to the atmosphere.

Author´s response: Thank you very much for your suggestion. We have addressed this issue and reconstructed the phrase. Please see line 773-774.

Line 743. Change “troubles” to “problems”

Author´s response: Thank you for your suggestion. The change was done. Please see line 776.

Line 744. Remove “into” in front of “have gone”. Change “de” to “a”. Remove dot after “biodegradability”and put a commar.

Author´s response: Thank you very much for your suggestion. We have addressed this issue and reconstructed the phrase. Please see line 777.

Line 765. Put “such” in front of “as”.

Author´s response: Thank you for your suggestion. The change was done.

Line 766. Remove “of great interest”

Author´s response: Thank you for your suggestion. The phrase was removed.

Line 775. I suggest removing “social problem”, because then you need explain it .

Author´s response: Thank you for your suggestion. The changed was done.

Reviewer 3 Report

The manuscript entitled:‘ Systematic transition from plastics to a new generation of renewable and recyclable bioplastics’, is interesting and scientifically relevant. Authors claim that the article aims to summarize the historical relevance of plastic materials, their adoption and evolution, current shortcomings, and the emerging trends of bio-based plastic material manufacture and implementation. The environmental effects of conventional and emerging plastics are presented, as well as mitigation strategies and the main applications of recent alternative materials. Challenges and opportunities for these biological plastics are discussed, and relevant business cases are presented in the context of modern needs of plastic materials. The relevant legislation that concerns plastic production, certification and the regulation of its disposal is also briefly discussed.

In my opinion, the section on the history of polymers is a quick and sufficient overview of the most important achievements in the field of plastics over the years. However, I missed information about one of the most durable plastics - Kevlar and Nomex, and one of the most famous women in the world of polymers - Stephanie Kwolek. Moreover, there is the lack of information about polyurethanes, the polymer taking the 5th place in the ranking of plastics with the highest global consumption.

At the manuscript, the environmental effects of conventional and emerging plastics are presented. The macro- and microplastics and also other plastic-derived pollutants and their impact on environment were described, which is significant. Moreover, challenges and opportunities of biodegradable plastics derived from its production to degradation and further prospective under circular economy are described. Nevertheless, there is the lack of summary of the possibilities of bio-plastic waste management and the possibility of their recycling / biodegradation with the conditions, products and their impact on the environment. I believe that such a summary (at a table form, in example) will definitely increase the scientific value of the review work.

The manuscript raises an important topic and therefore I recommend the work for publication after major revision.

Specific comments:

  1. Introduction, page 2, line 46 – ‘….terephtalate (PET)….’ – it should be written: ‘…poly(ethylene terephtalate) (PET)…’.
  2. Figure 2, page 5 – the content of some texts is indistinct and difficult to read, mainly because the font is too small.
  3. Point 3.4. ‘Other plastic-derived pollulants’ – there should be ‘Other plastic-derived pollutants’.
  4. Table 2, page 15 – ‘Static production in frasks’ – there should be ‘Static production in flasks’.

Please check the entire manuscript text for typos.

Author Response

We thank you for the detailed review and useful comments. We have done extensive language editing throughout the manuscript. We have followed your suggestion and accordingly revised the manuscript. The changes made in the revised manuscript are highlighted in GREEN color and shown in this document.  Please note below our responses to your comments.

The manuscript entitled:‘ Systematic transition from plastics to a new generation of renewable and recyclable bioplastics’, is interesting and scientifically relevant. Authors claim that the article aims to summarize the historical relevance of plastic materials, their adoption and evolution, current shortcomings, and the emerging trends of bio-based plastic material manufacture and implementation. The environmental effects of conventional and emerging plastics are presented, as well as mitigation strategies and the main applications of recent alternative materials. Challenges and opportunities for these biological plastics are discussed, and relevant business cases are presented in the context of modern needs of plastic materials. The relevant legislation that concerns plastic production, certification and the regulation of its disposal is also briefly discussed.

In my opinion, the section on the history of polymers is a quick and sufficient overview of the most important achievements in the field of plastics over the years. However, I missed information about one of the most durable plastics - Kevlar and Nomex, and one of the most famous women in the world of polymers - Stephanie Kwolek. Moreover, there is the lack of information about polyurethanes, the polymer taking the 5th place in the ranking of plastics with the highest global consumption.

Author´s response: Thank you for this insight; a brief comment on the historic relevance of Kevlar and Nomex, and the role of Stephanie Kwolek on the development of Kevlar, has been added. The historic position of polyurethane and its first inception are also included in this section. Please see line 130-134.

At the manuscript, the environmental effects of conventional and emerging plastics are presented. The macro- and microplastics and also other plastic-derived pollutants and their impact on environment were described, which is significant. Moreover, challenges and opportunities of biodegradable plastics derived from its production to degradation and further prospective under circular economy are described. Nevertheless, there is the lack of summary of the possibilities of bio-plastic waste management and the possibility of their recycling / biodegradation with the conditions, products and their impact on the environment. I believe that such a summary (at a table form, in example) will definitely increase the scientific value of the review work.

Author´s response: Thank you for your comment. A new section titled “Strategies for plastic reinsertion and environmental impact mitigation has been added under section number 5 to summarize the current state of plastic waste recycling through chemical and biological means. Table 5 in this subsection also summarizes the state of the art on biodegradation strategies, describing the conditions used in each reference. 

The manuscript raises an important topic and therefore I recommend the work for publication after major revision.

Specific comments:

  1. Introduction, page 2, line 46 – ‘….terephtalate (PET)….’ – it should be written: ‘…poly(ethylene terephtalate) (PET)…’.

Author´s response: Thank you for your comment. This error has been corrected. Please see line 45.

  1. Figure 2, page 5 – the content of some texts is indistinct and difficult to read, mainly because the font is too small.

Author´s response: Thank you very much for your comment. Figure 2 has been split and font size has been raised to improve readability.

  1. Point 3.4. ‘Other plastic-derived pollulants’ – there should be ‘Other plastic-derived pollutants’.

Author´s response: Thank you very much for your comment. This error has been corrected. Please see line 343.

  1. Table 2, page 15 – ‘Static production in frasks’ – there should be ‘Static production in flasks’.

Author´s response: Thank you very much for your comment. This error has been corrected. 

Please check the entire manuscript text for typos.

Author´s response: Thank you very much for your comment.The entire article has been reviewed and corrected

Round 2

Reviewer 1 Report

The review paper has been considerably revised and it has a much better flow now.

Some minor corrections:

  • line 534 - Polybutylene succinate (PBS) - has been already abbreviated at line 453. The same is the case with Polyhydroxyalkanoates which was abbreviated at line 429 and re-abbreviated at line 453, 564. Please revise every abbreviation in the article.
  • table 3 - there are some terms that are not defined in the table footer (i.e. PHA), and the term Polyethylene was used as PE and P.E. also - please revise
  • lines 384 - 386 - please rephrase. This sentence contains errors and it is also hard to understand.
  • line 735 - please change the order, from "plastics biodegradable" to "biodegradable plastics" and correct "food sector" to "the food sector".
  • line 741 - 746 - this section needs a reference.
  • line 799 - 802 - this sentence needs to be separated into two separate sentences.

After the implementation of these minor technical corrections, the article can be published.

Author Response

We thank you for your additional comments. We have addressed the issues according to your suggestions. Changes made in the revised manuscript are highlighted in YELLOW color and shown in this document.  Please note below our responses to your comments.

Comments and Suggestions for Authors

The review paper has been considerably revised and it has a much better flow now.

Some minor corrections:

  • line 534 - Polybutylene succinate (PBS) - has been already abbreviated at line 453. The same is the case with Polyhydroxyalkanoates which was abbreviated at line 429 and re-abbreviated at line 453, 564. Please revise every abbreviation in the article.

Author´s response: Thank you for your valuable comment.  We have addressed this issue according to your suggestion.

  • table 3 - there are some terms that are not defined in the table footer (i.e. PHA), and the term Polyethylene was used as PE and P.E. also - please revise

Author´s response: Thank you for your constructive comment. We improved table 3 according to your suggestion.

  • lines 384 - 386 - please rephrase. This sentence contains errors and it is also hard to understand.

Author´s response: Thank you for your comment. The sentence was reconstructed. Please see line 384-386.

Oxo-degradable plastics have enjoyed widespread adoption recently due to the technical feasibility of their implementation in already established manufacturing processes.

  • line 735 - please change the order, from "plastics biodegradable" to "biodegradable plastics" and correct "food sector" to "the food sector".

Author´s response: Thank you for your comment. We have addressed this issue according to your suggestion. Please see line 735-736.

  • line 741 - 746 - this section needs a reference.

Author´s response: Thank you for your comment. The reference was added.

  • line 799 - 802 - this sentence needs to be separated into two separate sentences.

Author´s response: Thank you very much for your comment. The sentence has been split to improve readability. Please see line 799-802.

Biorefineries focus on the comprehensive use of different organic wastes such as food, agro-industrial waste, forestry and algae biomass is of great interest to generate new precursors to bioplastics. Through different strategies such as bioprocesses with micro-organisms, the reduction of the use of non-renewable products are notable

After the implementation of these minor technical corrections, the article can be published

Reviewer 3 Report

I accept manuscript in revised version.

Author Response

Dear Reviewer,

We thank you for your valuable comments and suggestions. The manuscript was improved according to your suggestions.